# A druggable conformational switch in the c-MYC transactivation domain

Dilraj Lama [1,6] ✉, Thibault Vosselman[1,6], Cagla Sahin [1,2], Judit Liaño-Pons [1], Carmine P. Cerrato [1], Lennart Nilsson [3], Kaare Teilum [2], David P. Lane[1], Michael Landreh [1,4] ✉ & Marie Arsenian Henriksson [1,5] ✉

The c-*MYC* oncogene is activated in over 70% of all human cancers. The intrinsic disorder of the c-MYC transcription factor facilitates molecular interactions that regulate numerous biological pathways, but severely limits efforts to target its function for cancer therapy. Here, we use a reductionist strategy to characterize the dynamic and structural heterogeneity of the c-MYC protein. Using probe-based Molecular Dynamics (MD) simulations and machine learning, we identify a conformational switch in the c-MYC amino-terminal transactivation domain (termed coreMYC) that cycles between a closed, inactive, and an open, active conformation. Using the polyphenol epigallocatechin gallate (EGCG) to modulate the conformational landscape of coreMYC, we show through biophysical and cellular assays that the induction of a closed conformation impedes its interactions with the transformation/transcription domain-associated protein (TRRAP) and the TATA-box binding protein (TBP) which are essential for the transcriptional and oncogenic activities of c-MYC. Together, these findings provide insights into structure-activity relationships of c-MYC, which open avenues towards the development of shape-shifting compounds to target c-MYC as well as other disordered transcription factors for cancer treatment.

The gene encoding the c-MYC transcription factor is the most frequently deregulated oncogene across the majority of human cancers[1–3], with its overexpression and/or activation of downstream target genes in the vast majority of the cancer types analysed[4]. Deregulation of *c-MYC* can occur through elevated transcription of the gene, amplification, chromosomal translocation, point mutation, or protein stabilization, all of which release the otherwise tight control and increase expression levels of the protein in the cell[1,5]. Inhibition of c-MYC, on the other hand, has been observed to promote apoptosis, growth arrest, differentiation, senescence, metabolic changes, as well as tumour regression in several human cancer models, clearly indicating its potential as a target for anti-cancer therapeutics[6–9].

Targeting c-MYC using traditional structure-based drug design has remained challenging, due to its lack of a stable tertiary fold[10]. Intrinsically disordered proteins (IDPs) like c-MYC exist in an ensemble of conformational states, which facilitate their interaction with multiple binding partners. This feature is illustrated by the structural characterization of peptide derivatives from different regions of c-MYC[10]. The amino-terminal residues 13–30 form a fuzzy complex with the protein phosphatase 1 nuclear targeting subunits (PNUTS)[11]. Residues 55–68 predominantly adopt a polyproline II (PPII)

[1]Department of Microbiology, Tumor and Cell Biology (MTC), Karolinska Institutet, Biomedicum, SE-17165 Stockholm, Sweden. [2]Department of Biology, Structural Biology and NMR Laboratory and the Linderstrøm-Lang Centre for Protein Science, University of Copenhagen, DK-2200 Copenhagen, Denmark. [3]Department of Biosciences and Nutrition, Karolinska Institutet, SE-14813 Huddinge, Sweden. [4]Department of Cell- and Molecular Biology, Uppsala University, SE-751 24 Uppsala, Sweden. [5]Division of Translational Cancer Research, Department of Laboratory Medicine, Lund University, SE-221 00 Lund, Sweden. [6]These authors contributed equally: Dilraj Lama, Thibault Vosselman. ✉e-mail: dilraj.lama@ki.se; michael.landreh@ki.se; marie.arsenian.henriksson@ki.se

conformation in complex with the tumour suppressor MYC box-dependent-interacting protein 1 (BIN1)[12]. Downstream residues 96–111 form an amphipathic alpha-helix when bound into the groove formed by the TATA binding protein (TBP) and the amino-terminal domain of TBP-associated factor 1 (TAF1[TAND1])[13]. A peptide fragment (residues 260–267) from the central region binds the chromatin-associated tryptophan-aspartic acid (WD) repeat-containing protein 5 (WDR5) in an extended conformation by occupying a shallow hydrophobic cleft on the surface of the protein[14]. Importantly, the carboxy-terminal region (residues 351–439) of c-MYC dimerizes with its obligate binding partner MYC-associated factor X (MAX) for DNA recognition by adopting a basic-helix-loop-helix leucine-zipper (bHLHZip) conformation[15]. Furthermore, the peptides derived from c-MYC and the transformation/transcription domain-associated protein (TRRAP) have been reported to undergo a disorder to order transition in the formation of a structurally-stable complex[16]. In addition, the structure of residues 61–89 from MYCN (a member of the MYC family) has been revealed in complex with the protein kinase Aurora-A, in which the amino-terminus of the peptide adopts an elongated state followed by an alpha helix towards its carboxy-terminus[17]. Hence, the protein requires specific local conformations to exert its biological activity: in a dynamic ensemble, the disordered c-MYC would fluctuate between metastable "active" conformations for interacting with its binding partners, and "inactive", resting states not amenable for binding. Molecular interventions that can stabilize or propagate such "inactive" configurations would thereby have a significant impact on its activity. In fact, structural ensemble modulation of IDPs has been recognized as a useful strategy for their inhibition[18,19]. A comprehensive understanding of the structural features will be essential to design ligands that can efficiently drive the conformational transition of the target proteins. However, most high-resolution structure determination methods do not allow a detailed characterization of disordered protein states. Hence, a different conceptual framework is required for their structural exploration and targeting for therapy. This is exemplified by that despite continuous efforts, the only promising drug candidate in clinical trials to date is the OMOMYC peptide (clinicaltrials.gov identifier: NCT04808362)[20]. This mini-protein is derived from the bHLHZip domain of MYC with four rationally incorporated mutations that renders the peptide to preferentially bind to MAX or to itself and will thus modulate MYC-mediated transcription by inhibiting MYC:MAX dimerization and DNA-binding[20,21].

In this work, we have taken a distinct approach to address these challenges and developed a peptide-based computational strategy to identify a conformational switch in c-MYC. The transition samples different structural configurations which are directly related to the association of c-MYC with proteins required for its transcriptional activity, and thus present an opportunity for therapeutic intervention. We have further employed probe-based molecular dynamics (MD) simulations, mass spectrometry, and cell assays to demonstrate that targeting this conformational switch using a model compound inhibits interactions between c-MYC and its binding partners, establishing a potentially druggable feature.

## Results

### c-MYC harbors a conditionally folded switch

The high conformational flexibility of c-MYC makes it very challenging to structurally characterize the native full-length protein. Hence, we undertook a reductionist approach by dividing the protein into 50-residue non-overlapping peptides resulting in nine derivatives (MYC$_{1-50}$ to MYC$_{401-439}$, Fig. 1a). Except for the last peptide, which is 39 residues in length, all other peptides consist of 50 residues. We then subjected each peptide to all-atom MD simulations with ff19SB force-field and OPC water model (see Supplementary Information for details), and calculated the radius of gyration ($R_g$) for the structures generated through their independent microsecond simulations to create a probability distribution profile (Fig. 1b, black curves). Analysis of these plots shows that the peptides can be classified into two major groups based on their nature and scale of $R_g$ distribution. The group of the five peptides spanning amino acids 1–50, 51–100, 151–200, 351–400, and 401–439 has narrow normal distributions roughly centered around a mean of 15 Å, suggesting that they are mostly compact and have a low degree of fluctuation. However, peptides covering residues 101–150, 201–250, 251–300, and 301–350 display variable types of distributions exhibiting a wider range of $R_g$ values up to 35 Å (Fig. 1b, black curves). This indicates that these peptides have a greater flexibility than the other parts of the protein and sample relatively more extended states. Overall, the peptides derived from the c-MYC protein exhibit variable sampling properties.

Mixed-solvent MD simulations of proteins is a technique applied in structure-based drug discovery for the identification of cryptic ligand binding pockets, fragment screening, location of hotspot interactions, and assessment of druggability[22,23]. We therefore chose to explore the influence of additives on the conformational distributions of the nine different c-MYC peptides. We selected benzene as the co-solvent as it is one of the most commonly used small organic probes in such simulations[24,25]. All the mixed-solvent MD simulations were initiated from the same starting structure in pure water with the addition of 0.4 M benzene. By comparing the resulting $R_g$ distributions with water-only based simulations, we mostly observed profiles that are qualitatively very similar with moderate variations between the two solvent conditions (Fig. 1b, red curves). The only exception is the peptide covering residues 101–150. Here, the presence of the benzene probe has a significant impact on the conformational sampling. This region undergoes a pronounced transformation from a wide $R_g$ distribution range from 5 to 35 Å to a narrow range between 5 and 15 Å with the mean at 12 Å. This clearly indicates that the peptide is much more compact in the presence of benzene. The benzene probe-based peptide screening of c-MYC was further expanded with eight additional 50-residue derivatives (MYC$_{25-74}$ to MYC$_{375-424}$) that overlap with the initial library of nine peptides (MYC$_{1-50}$ to MYC$_{401-439}$; Supplementary Fig. 3a). The conformational distributions of the overlapping peptides indicate that they show relative transition towards a compact state to different degrees with the addition of benzene (Supplementary Fig. 3b), but none of them undergo a significant ensemble modulation between the two environments as was observed for MYC$_{101-150}$.

Intrinsically disordered proteins (IDPs) are generally characterized by amino-acid compositions with high proportions of hydrophilic and charged residues. To understand the divergent behavior of the region between residues 101 and 150, we calculated the disorder and hydrophobicity scores of c-MYC. Notably, and as expected, we found an inverse correlation between the two variables, i.e., a high percentage of c-MYC observed in disordered regions correlating with hydrophilic stretches (Fig. 1c). Interestingly though, the peptide spanning amino acids 101–150, which is predicted to be ordered is also the most hydrophobic segment in the protein. This observation is corroborated with multiple algorithms, albeit with different scales (Supplementary Fig. 4). A closer examination shows that 50% of the residues in this region are hydrophobic and bulky. This analysis highlights that the region is an epitope within the disordered c-MYC, which does not possess a prototypical IDP sequence composition. We explored a rationale for the conformational divergence induced by benzene by computing the enthalpic energy (E) of all the simulated structures in combination with their corresponding $R_g$ values. To this end, we generated a two-dimensional "$R_g$-E" distribution profile for residues 101–150 (Fig. 1d). It is evident from this plot that compact conformations are distributed along the energy scale, ranging from favorable (low energy score) to unfavorable (high energy score), while extended conformations are associated predominantly with unfavorable states. The bulky hydrophobic residues tend to shield themselves from the polar environment, and hence drive the formation of the

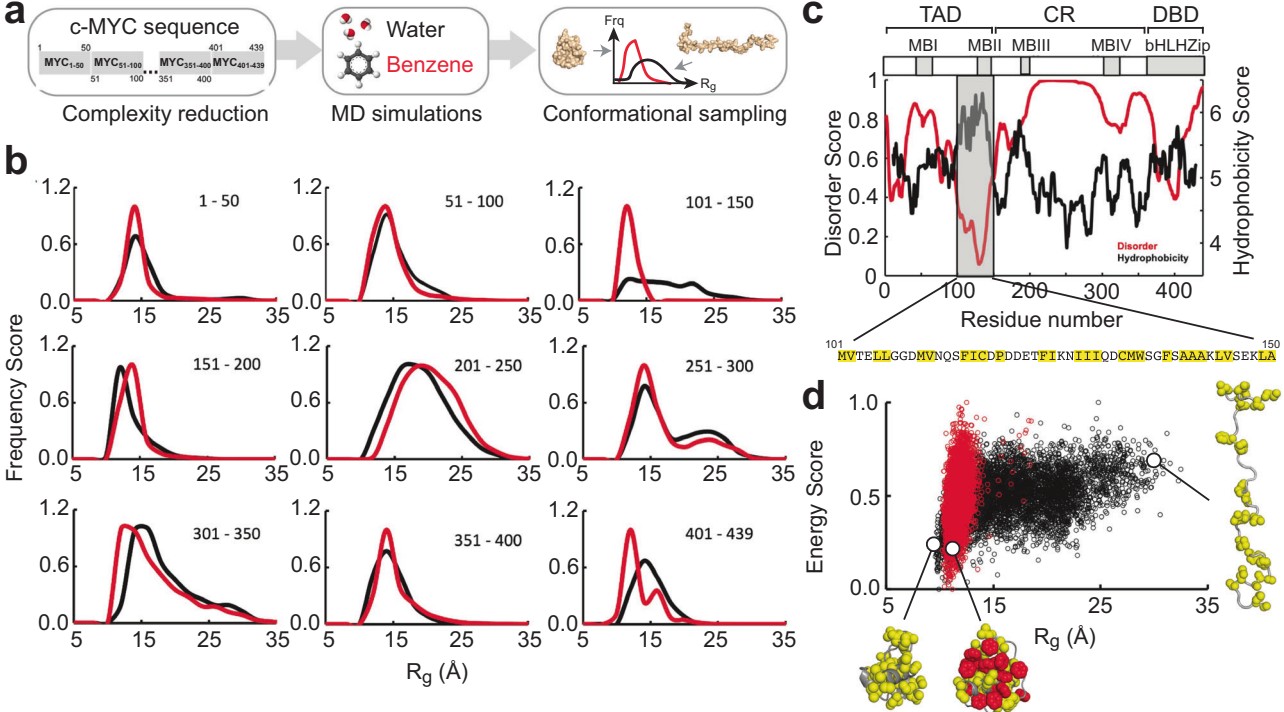

**Fig. 1 | Identification of a conformational switch in c-MYC. a** Overview of the MD strategy employed to identify regions that exhibit probe-induced conformational changes. **b** Normalized frequency distribution of $R_g$ computed from the ensemble of structures generated from explicit solvent MD simulations in water (black) and water + benzene (red) for all nine c-MYC derived peptides as indicated. **c** Disorder (VSL2 algorithm) and hydrophobicity (Miyazawa hydrophobicity scale) profile of c-MYC. The modular architecture (TAD: Transactivation Domain, CR: Central Region, DBD: DNA Binding Domain) of c-MYC, MYC Boxes I–IV (MBI – IV) and the bHLHZip domain is shown at the top. The hydrophobic residues between positions 101 and 150 are highlighted in yellow background. **d** $R_g$-E (energy normalized in the scale from 0 to 1) distribution plots of coreMYC in water (black) and water + benzene (red) solvents. Representative models of the compact and extended states of coreMYC from the ensemble of MD generated structures are indicated. The hydrophobic residues and the benzene probes are shown as yellow and red spheres, respectively.

hydrophobic collapsed state of the peptide. The addition of benzene completely abolishes the extended configurations of the peptide. This change is due to interactions between benzene and the hydrophobic residues, acting as a molecular "glue" that drives the equilibrium towards the compact state (Fig. 1d). A comparative analysis of similar "$R_g$-E" scatter plots, along with the time-dependent evolution of $R_g$ across all the peptide derivatives of c-MYC show that except for a notable extent in $MYC_{75-124}$, which overlaps with $MYC_{101-150}$, the addition of benzene does not induce a similarly pronounced transition elsewhere in the protein (Supplementary Figs. 5 and 6). This distinct feature of the epitope was further examined by subjecting the peptide to mixed-solvent simulations with three different chemical probes, which include propane (hydrophobic: aliphatic), methanol (hydrophilic: h-bond donor), and acetaldehyde (hydrophilic: h-bond acceptor). We find that propane shifts the structural ensemble of the peptide towards a compact conformation as observed for benzene (hydrophobic: aromatic), while the hydrophilic probes (methanol and acetaldehyde) marginally increase the population of the compact state, but have no significant impact on the epitope (Supplementary Fig. 7). These data further underscores the distinct sequence composition of the region within c-MYC, which makes it particularly susceptible to conformational modulation by certain molecular probes. We conclude that the peptide covering residues 101–150, which we refer to as "coreMYC" (COnformational REgulator of c-MYC), is susceptible to ligand-induced conformational changes.

## EGCG binding induces compaction of coreMYC

The finding that the benzene probe induced compaction of coreMYC in MD simulations led us to explore more soluble and physiologically

viable small molecules that could induce similar conformational shifts but are better suited for experimental validation. We selected Epigallocatechin gallate (EGCG), a polyphenolic compound that is abundant in green tea (Fig. 2a). Due to its high biocompatibility, EGCG is considered a pro-drug with anti-cancer activity[26]. It is widely employed as a non-specific inhibitor of protein aggregation[27] and has more recently been shown to modulate the conformational ensembles of IDPs[28], including the disordered transactivation domain (TAD) of p53[29].

To investigate whether EGCG could elicit similar effects as the benzene probe, we turned to Ion Mobility spectrometry in combination with native Mass Spectrometry (IM-MS). In native MS (nMS), protein complexes are gently transferred from near-physiological solutions to the gas phase using electrospray ionization (ESI), preserving their three-dimensional fold and non-covalent interactions. Mass measurements of the desolvated, intact complexes can then be employed to determine the stoichiometries of their components. With ion mobility separation (IM), the complexes are separated in a gas-filled cell according to their collision cross section (CCS), which is analogous to the $R_g$, although the charging and desolvation processes can cause additional conformational changes[30]. By combining IM with nMS, we can monitor the conformational states of individual protein populations, e.g., for protein-ligand complexes with varying stoichiometries. nMS of recombinantly produced coreMYC indicated the presence of the peptide in its monomeric form with partial cleavage of the amino-terminal methionine as well as 1–4 sodium adducts (Fig. 2b). Ion mobility analysis revealed the presence of several populations, representing extended, compact, and intermediate conformations, with the compact state, composed of two populations, being the least abundant (Fig. 2c). We then added increasing concentrations of EGCG

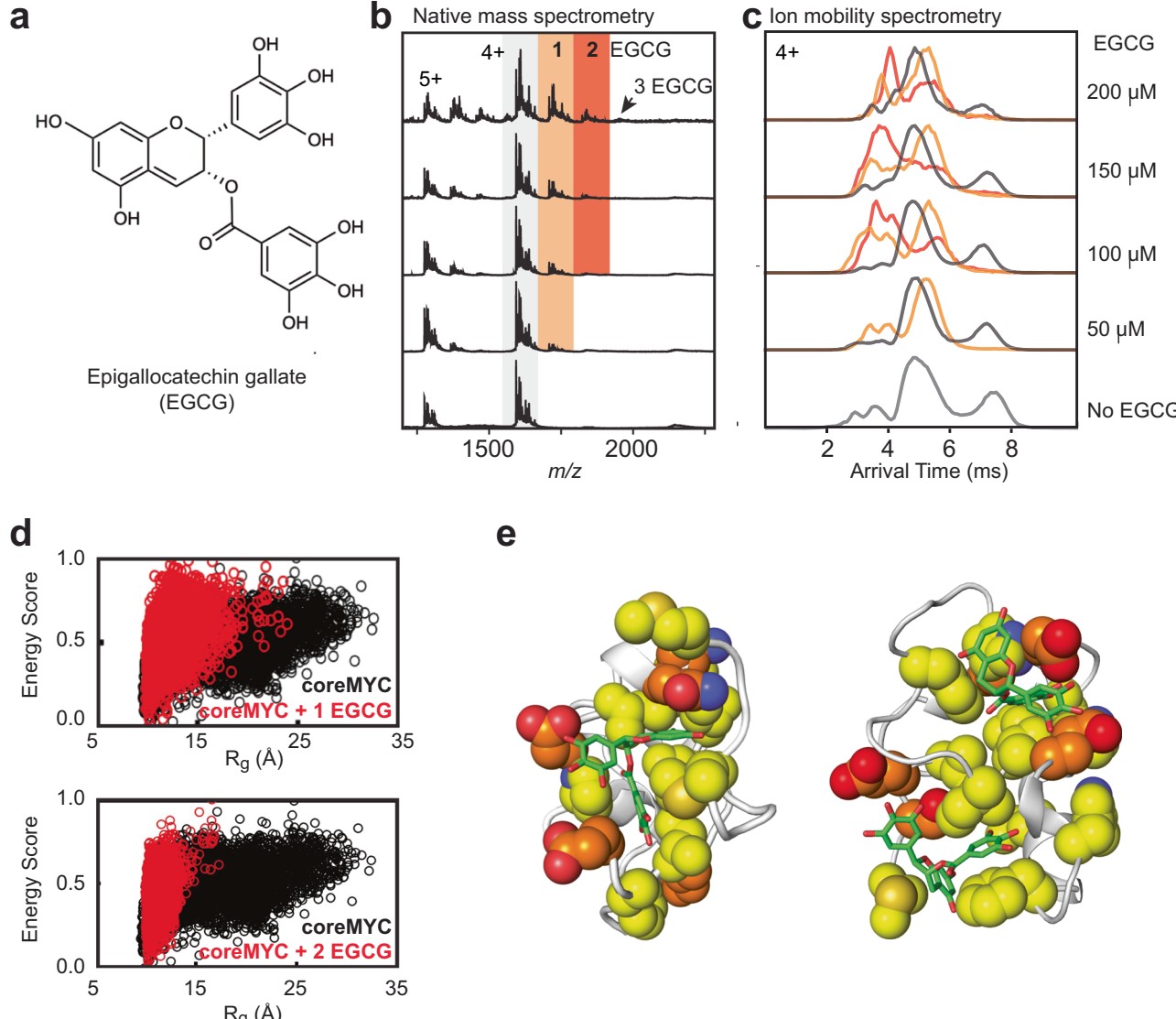

**Fig. 2 | Characterization of the interaction between coreMYC and EGCG. a** The chemical structure of EGCG. **b** Native mass spectra of coreMYC in the presence of increasing concentrations of EGCG. The observed charged states and number of EGCG molecules bound to coreMYC are indicated. **c** Arrival time distributions for the 4+ charge state of coreMYC in different concentrations of EGCG. **d** $R_g$-E (energy normalized in the scale from 0 to 1) distribution plots of coreMYC generated from MD simulations of the peptide with one (upper panel) or two (lower panel) EGCG molecules (red), respectively. $R_g$-E distributions of coreMYC in water (black) is shown for reference. **e** Representative structures from MD simulations of coreMYC bound to one (left) or two (right) EGCG molecules, respectively. The residues from coreMYC with contact score >1 with EGCG are shown in spheres and the EGCG is depicted in stick representation (green). See Supplementary Figs. 9e, f and 10e, f for more details on the residues. Side-chain carbon atoms of hydrophobic residues are shown in yellow and hydrophilic residues in orange. Oxygen and nitrogen atoms are shown in red and blue respectively.

and monitored the interaction by nMS. Binding of one EGCG molecule to coreMYC was readily observed at an EGCG concentration of 50 µM. At concentrations of 100 µM and above, we found two EGCG adducts with the third adduct observed at the highest tested concentration of 200 µM (Fig. 2b). Plotting the signal intensity of the 1:1 complex as a function of EGCG concentration showed no apparent saturation within the concentration regime that could be tested by nMS, indicating non-specific binding (Supplementary Fig. 8a)[31]. Strikingly, IM-MS analysis of the complexes between EGCG and coreMYC showed a strong shift in conformations from zero to two bound molecules; binding of one single EGCG compound almost completely depleted the most extended population, while binding of two EGCG molecules resulted in an almost exclusively compact state (Fig. 2c). For the 5+ charge state, we furthermore observed an increase in compact coreMYC without bound EGCG, which indicates that some of the peptide-ligand

complexes dissociate during the transfer to the gas phase (Supplementary Fig. 8b).

To gain more insights into the structural effects of EGCG on coreMYC, we conducted MD simulations of the peptide in the presence of one or two EGCG molecules. The $R_g$-E distribution plots from the simulations clearly show that the peptide is predominantly in the compact conformation (Fig. 2d), analogous to the benzene mixed-solvent simulations (Fig. 1d). This conformational landscape is consistently observed in both replicates for each of the individual systems (Supplementary Figs. 9a, b, 10a, b and 11). To directly correlate these findings with experimental data, we computed the theoretical CCS values for the compact states of coreMYC in the MD simulations using IMPACT[32] and compared them to the experimentally determined $^{TW}CCS_{N2}$ values of coreMYC:EGCG complexes. Indeed, the experimental $^{TW}CCS_{N2}$ values of 820, 825, and 897 Å$^2$ are in reasonable

agreement with the theoretical CCSs of 820, 881, and 923 Å$^2$, respectively for zero, one, or two EGCG molecules bound. We additionally sought to validate the effect of EGCG on coreMYC using solution nuclear magnetic resonance (NMR) spectroscopy. However, the chemical shift perturbations observed in coreMYC during EGCG titrations could not be separated with sufficient confidence from potential titration artefacts. The data has therefore not been included in this study.

Lastly, to obtain a detailed model for EGCG-induced compaction of coreMYC, we compared the IM-MS data to the contacts between EGCG and coreMYC from simulation trajectories. The MD data indicate a dynamic mode of association with multiple residues involved in the interaction. However, the specific residues that are involved in the recognition are not exactly identical and could vary between replicates (Supplementary Figs. 9c–f and 10c–f). This indicates (as was also deduced from the nMS measurements) that EGCG associates with coreMYC in a non-specific manner, which is further highlighted with the multiple bound conformations observed from clustering based on the residues that form maximum contact during the MD simulations (Fig. 2e; Supplementary Figs. 9g, h and 10g, h). It is interesting to note that in the 1:1 complex, EGCG is wedged into an almost cavity-like structure, but associates in distinct regions of the peptide in the 1:2 complex. These conformations are observed to be relatively stable (Supplementary Fig. 12), which indicate that they are potential representatives of the multiple metastable complexes formed during the dynamic interaction between EGCG and coreMYC. In summary, the combination of MS, and MD provide compelling evidence at the molecular level that EGCG associates with coreMYC in a heterogenous manner leading to the induction of a compact conformation.

## Conformational switching of coreMYC regulates access to MYC Box II

The c-MYC protein is characterized by the presence of at least four highly evolutionary conserved sequence motifs termed MYC Box I – IV (MBI – IV), where MYC Box II (MBII; residues 128–143) is present within coreMYC (Fig. 1c). This motif serves as a binding site for multiple proteins that regulate the transcriptional activity of c-MYC[2,33,34]; yet how these interactions are controlled in the disordered c-MYC is not known. We therefore asked whether the conformational transitions of coreMYC between compact and extended states could translate into regulating the interactions of MBII with its binding partners. For this purpose, we created a dataset of eighteen direct interactors of c-MYC (Supplementary Table 1), which have been experimentally characterized to recognize the protein either through MBII or other parts of the amino-terminal region which includes coreMYC[33,34]; and turned to AlphaFold2 (AF2), which besides predicting highly accurate protein structures[35,36], has also been demonstrated to be an excellent tool for peptide: protein docking[37,38]. We examined its utility by predicting complex structures from sequences from all structurally resolved c-MYC derived peptides bound to different interactors (Supplementary Fig. 13a–e). The accuracy of the modeled complexes varied to different degrees (ipTM score 0.3–0.9), however and more importantly for our study, the conformation (measured in $R_g$) of the interacting c-MYC peptides showed a good agreement for all complexes between the experimental and the AF2 structures (Supplementary Fig. 13f). Based on this benchmarking exercise, we then modeled coreMYC in complex with the eighteen selected partner proteins (Supplementary Fig. 14) and computed the $R_g$ of the peptide. Comparing the conformational distributions of coreMYC in each complex and in MD simulations shows that the peptides have a notably higher $R_g$ when bound to a partner protein (Fig. 3a). Inspection of the residue-wise binding energy contribution involved in the predicted complexes further demonstrated that coreMYC interacts preferentially via hydrophobic residues (Fig. 3b), and the compaction of coreMYC is

driven by hydrophobic collapse (Fig. 1d). This finding suggests that the peptide has to adopt an extended, open conformation to regulate access to MBII in order to recognize its binding partners.

To test this observation experimentally, we selected the adapter protein TRRAP, a critical and well characterized cofactor of c-MYC[34,39]. The minimal binding region of TRRAP with c-MYC (residues 2033–2088) has been described to undergo a disorder-to-order transition when complexed with MBII[16]. We investigated the complex formation of coreMYC and a peptide from TRRAP spanning amino acids 2038–2087 (TRRAP$_{2038–2087}$) using nMS. Under near-physiological solution conditions, both peptides have well-resolved spectra showing only monomeric species (Fig. 3c). Mass spectra of an equimolar amount of coreMYC and TRRAP$_{2038–2087}$ show peaks corresponding in mass to a 1:1 complex (Fig. 3c; Supplementary Fig. 15). Importantly, no other stoichiometries were observed, suggesting a specific association. To test the effect of EGCG on this interaction, we briefly incubated coreMYC with 150 μM EGCG prior to the addition of TRRAP$_{2038–2087}$ and recorded native mass spectra. Strikingly, the presence of EGCG completely inhibited complex formation. Instead, we observed binding of up to two EGCG molecules to both peptides (Fig. 3c). We conclude that association with EGCG inhibits the ability of coreMYC to interact with TRRAP$_{2038–2087}$.

To elucidate whether the inhibitory effect of EGCG is due to compaction of coreMYC, we analysed the AF2 predicted structures of coreMYC:TRRAP$_{2038–2087}$ complex. The AF2 model suggests an elongated and predominantly helical structure of the two peptides in the complex state (Supplementary Fig. 16). Interestingly, similar peptide derivatives from c-MYC and TRRAP has been shown to form a structurally stable complex with high helical content[16]. The highest ranked model from the predicted structures was then subjected to multi-copy MD simulations. Binding energy calculations from the replicates show that the hydrophobic residues in both peptides contribute the majority of energetically favorable interactions (Fig. 3d, e; Supplementary Fig. 17), indicating that the association is predicted to be driven by hydrophobic interaction. Importantly, residue W135 from coreMYC, which has been reported to be involved in the recognition of TRRAP[16], is part of the hydrophobic interaction network in the modeled complexes (Fig. 3d, e; Supplementary Fig. 17). We also created in silico glycine mutant derivatives of W135 as well as other hydrophobic residues in the MBII motif of coreMYC and generated AF2 models of their complexes with TRRAP$_{2038–2087}$. The binding energy computed from the highest ranked models indicate that the potency of interaction with TRRAP$_{2038–2087}$ decreases with the replacement of hydrophobic residues (Supplementary Fig. 18). Furthermore, from MD simulations, we conclude that coreMYC engages in nonspecific interactions with EGCG (Fig. 2e; Supplementary Figs. 9e–h and 10e–h) that reduce its ability to interact with TRRAP$_{2038–2087}$.

Next, having observed that EGCG inhibits complex formation of coreMYC with the disordered TRRAP$_{2038–2087}$ peptide, we also investigated its impact on the binding of the epitope to a folded protein. For this purpose, we selected TBP, an integral component of the multimeric transcription factor IID (TFIID) protein assembly[40]. nMS of full-length, human TBP showed the formation of 1:1 complexes with coreMYC (Fig. 3f), as anticipated from structural and biochemical studies, which have shown direct association of residues 96–125 (the amino terminal half of coreMYC) with a binary complex of TBP and TAF1[13]. Pre-incubation of coreMYC with a 10-fold excess of EGCG almost completely abolished complex formation, confirming that EGCG indeed interferes with the interaction by affecting coreMYC directly. A modeled complex of coreMYC with human TBP (Fig. 3g) shows that the recognition occurs in an extended conformation predominantly through the hydrophobic residues present in this region (Fig. 3h; Supplementary Fig. 19), which are also involved in EGCG binding and compaction of coreMYC (Fig. 2d, e and Supplementary Figs. 9–11).

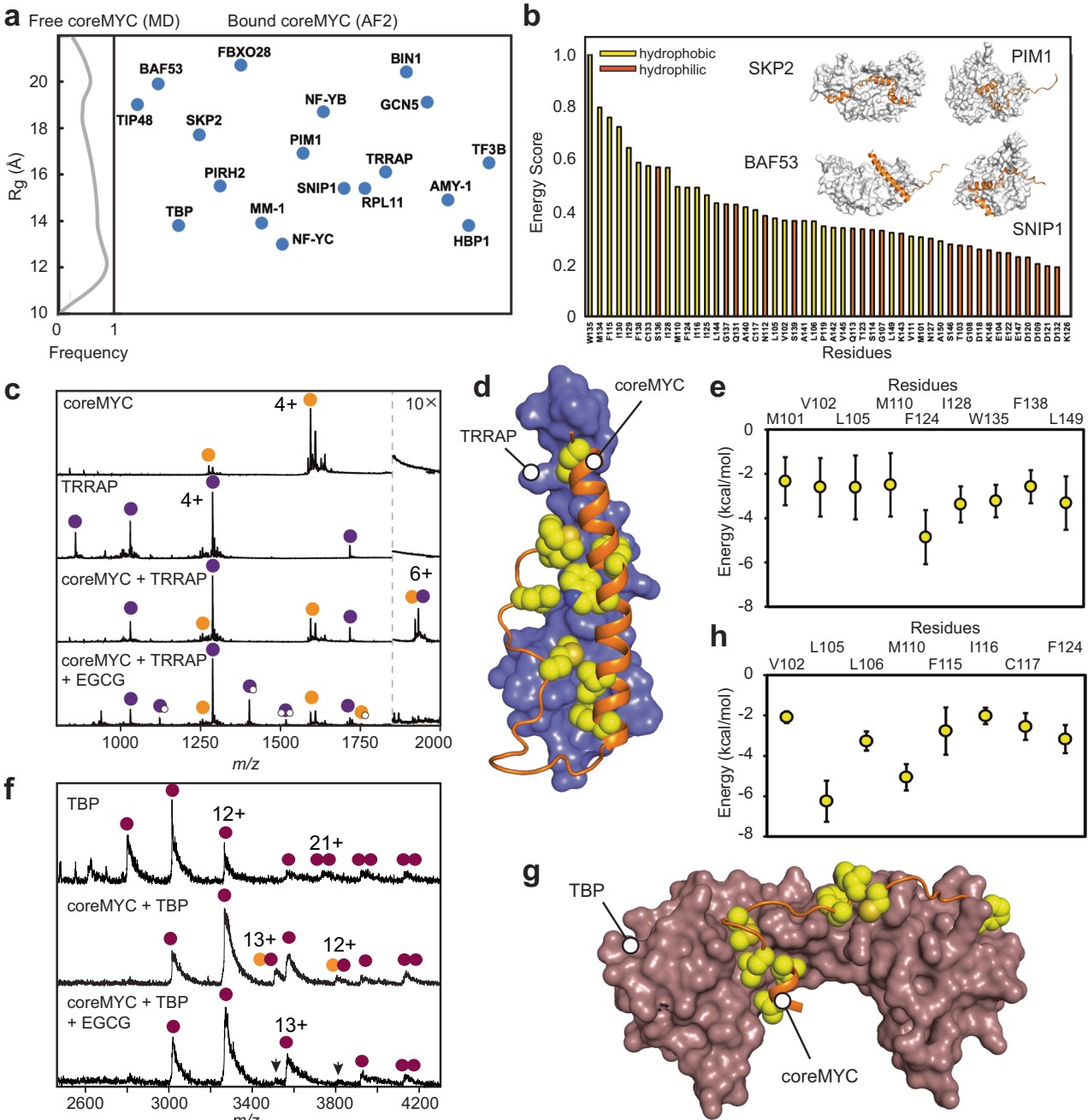

**Fig. 3 | Active conformation of coreMYC and inhibition of coreMYC interactions. a** $R_g$ values of coreMYC computed from AF2 modeled complexes with its binding partners mapped with the $R_g$ distribution obtained from MD simulations (insert to the left). **b** Cumulative residue-wise binding energy score (normalized in the scale from 0 to 1) of coreMYC in the bound complexes. Representative AF2 modeled complexes are shown with coreMYC rendered as orange cartoon and the binding partners in gray surface (see Supplementary Fig. 14 for the complete set of modeled complexes). **c** Native mass spectra of coreMYC (orange circles), TRRAP$_{2038-2087}$ (violet circles), coreMYC with TRRAP$_{2038-2087}$, and coreMYC with TRRAP$_{2038-2087}$ after pre-incubation with EGCG. The zoom region from $m/z$ 1800–2000 shows the 1:1 coreMYC:TRRAP$_{2038-2087}$ complex formed in the absence of EGCG. **d** Representative structure of the complex between coreMYC (orange ribbon) and TRRAP$_{2038-2087}$ (blue surface) from MD simulations initiated from the AF2 generated model. Residues from coreMYC that contribute significantly to binding ($\leq -2$ kcal/mol) are rendered as yellow spheres. **e** Residue-wise binding energy contribution from coreMYC. Only residues with energetic contribution $\leq -2$ kcal/mol is shown. Data are represented as Average ± S.D. computed with $n = 1000$ simulated structures. See Supplementary Fig. 17 for the complete set of energetic contribution from both peptides. **f** Native mass spectra show that human full-length TBP (brown circles) forms a 1:1 complex with coreMYC (orange circles), Pre-incubation of coreMYC with EGCG suppresses complex formation (arrows). **g** Representative model from MD simulation of the first 24 residues from coreMYC (residues: 101–124, orange ribbon) and the core domain of human TBP (brown surface). Residues from coreMYC that contribute significantly to binding ($\leq -2$ kcal/mol) are rendered as yellow spheres. **h** Residue-wise binding energy contribution from coreMYC. Data are represented as Average ± S.D. computed with $n = 1000$ simulated structures. Only residues with energetic contribution $\leq -2$ kcal/mol is shown. See Supplementary Fig. 19 for the complete set of binding energy contribution from coreMYC.

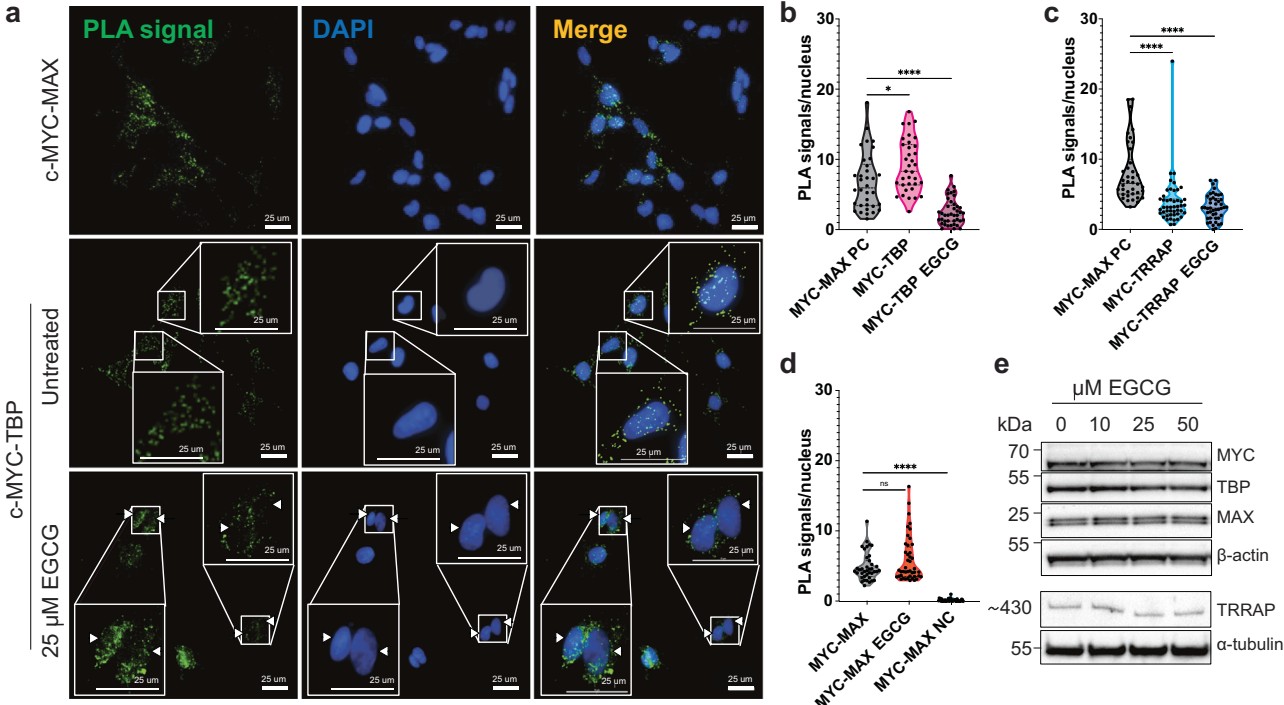

**Fig. 4 | EGCG inhibits interactions of c-MYC with TBP and TRRAP in cancer cells.**
**a** Representative images of isPLA for c-MYC in SH-SY5Y cells from $n = 3$ biologically independent experiments. Top row: PLA positive control of c-MYC: MAX complex (green) co-localized with nuclear DAPI staining (blue). Middle row: PLA signal for c-MYC: TBP interaction (green) and nuclei (blue) in untreated cells. Insert: Enlargement of representative cells showing nuclear PLA signal (arrow heads). Bottom row: PLA for c-MYC: TBP complex (green) in cells treated with 25 μM EGCG shows a reduced nuclear signal for the complex formation compared to control. Insert: Enlargement of representative cells with weak or no nuclear PLA signal (arrow heads). Yellow indicates merge between respective PLA signal with DAPI. Scale bar is 25 μm. **b** Quantification of nuclear c-MYC:TBP signal ($n \geq 10$ samples/biological experiment) in untreated and 25 μM EGCG treated cells showing reduced PLA signal compared to control; $n = 3$ biologically independent experiments. MYC-MAX (positive control, PC) *versus* MYC-TBP 0 μM EGCG $p = 0.0190$; PC versus MYC-TBP 25 μM EGCG $p = <0.0001$. **c** Quantification of nuclear c-MYC:TRRAP signal in untreated and 25 μM EGCG treated cells showing reduced PLA signal compared to

control; $n = 3$ biologically independent experiments. PC *versus* MYC-TRRAP 0 μM EGCG $p = <0.0001$; PC *versus* MYC-TRRAP 25 μM EGCG $p = <0.0001$.
**d** Quantification of nuclear c-MYC:MAX signal in untreated and EGCG-treated cells shows that interaction is not reduced with 25 μM EGCG; $n = 3$ biologically independent experiments. MYC-MAX 0 μM EGCG *versus* MYC-MAX 25 μM EGCG $p = 0.0882$ (n.s.), *versus* MYC negative control (MYC NC) $p = <0.0001$. n.s. = not significant. $*p = <0.05$ and $****p = <0.0001$ (Statistics for **b**–**d** were run using one-way ANOVA, Dunnett's multiple comparison test with a single pooled variance). **e** Western blot analysis shows that MYC, TBP, TRRAP, and MAX levels are unaffected by EGCG treatment. β-Actin is used as loading control for MYC, TBP, and MAX, and α-tubulin for TRRAP. Molecular weight markers (kDa) are indicated to the left. One representative blot is shown out of $n = 3$ biologically independent experiments (quantification is presented in Supplementary Fig. 21d). For western blot and PLA assays, the samples were derived from the same biological replicate and processed in parallel.

Based on these findings, we conclude that the extended state of coreMYC with exposed hydrophobic residues constitutes an active recognition module that enables coreMYC interaction with its binding partners, while the compact conformation represents an inactive state of the peptide.

## EGCG prevents coreMYC dependent recruitment of TBP and TRRAP in cancer cells

Having established that the conformation of coreMYC regulates binding of target proteins and can be modulated by EGCG, we sought to explore the basic mechanism in a cellular context. To this end, we monitored the protein–protein interaction (PPI) of c-MYC with TBP and with TRRAP using in situ proximity ligation assays (isPLAs) in the human neuroblastoma cell line SH-SY5Y, which constitutively over-expresses c-MYC and is a well-established model for c-MYC activity in tumorigenesis[41,42].

The isPLA showed a distinct signal for c-MYC:TBP complexes predominantly in the nucleus but also in the cytoplasm (Fig. 4a). Next, we analysed whether this complex formation could be disrupted by exposure to EGCG. To this end, cells were treated with EGCG concentrations ranging from 0 to 100 μM over a 24 h period. The SH-SY5Y cells did not display any significant change in viability when treated

with up to 25 μM EGCG, beyond which we observed a dose-dependent decrease in survival, as judged by live-cell imaging (Supplementary Fig. 20a). Similarly, analysis with the WST1 assay showed only a minor decrease in metabolic activity in the presence of 20 μM EGCG (Supplementary Fig. 20b). These findings indicate that although higher concentrations of EGCG are toxic to the cells, treatment with 25 μM EGCG is well-tolerated. We therefore focused on the signals from the isPLA assay in the presence of 25 μM EGCG, and found a substantial loss of PLA signals from nuclear c-MYC:TBP complexes at this EGCG concentration as compared to untreated cells (Fig. 4b). We next analyzed whether EGCG could also disrupt the complex between c-MYC and TRRAP. Our data show that a small fraction of TRRAP was engaged in a direct nuclear interaction with c-MYC, and this binding was inhibited in the presence of 25 μM EGCG (Fig. 4c). Lastly, we examined whether EGCG would also affect the carboxy-terminal bHLHZip interaction between c-MYC and MAX. Importantly, we found that treatment with EGCG did not affect the amount of nuclear PLA signals for c-MYC-MAX complexes, supporting the idea that EGCG predominantly affects interactions involving MBII (Fig. 4d).

The robustness of the assay was also demonstrated in control experiments where no PLA signal was observed in the negative control (with only c-MYC antibody; Supplementary Fig. 21a), while the positive

control, measuring the interaction between c-MYC and MAX, showed a strong signal inside the nuclei in the absence of EGCG (Fig. 4a). Quantification of the PLA signals showed a dose-dependent reduction of nuclear c-MYC:TBP and c-MYC:TRRAP complexes with increasing EGCG concentration (Fig. 4b, c; Supplementary Fig. 21). In order to verify that the reduction in PLA signals was not due to reduced protein levels in the cell, we analysed the expression of c-MYC, MAX, TBP, and TRRAP after EGCG incubation using Western blot. Cells were treated with 0–50 μM EGCG and protein expression was analyzed after 24 h (same conditions as for the isPLA). We found that treatment with 25 μM EGCG did not affect expression of any of the studied proteins (Fig. 4e; Supplementary Fig. 21d). Furthermore, EGCG did not affect *c-MYC* mRNA levels (Supplementary Fig. 21e). Thus, these data confirm that the observed reduction in PLA signals is due to inhibition of interaction between c-MYC:TBP and c-MYC:TRRAP and not due to reduced protein expression. Taken together, our findings indicate that ECGC efficiently disrupts c-MYC:TBP and c-MYC:TRRAP complex formation in cancer cells.

## Discussion

Here, we have used a hybrid MD-MS strategy to identify a conformational switch region within the structurally protected amino-terminal TAD of c-MYC[43]. The peptide termed "coreMYC" includes the functionally important MBII motif and switches between an active, extended state and a compact, inactive configuration that can be induced by the biocompatible ligand EGCG. Our study provides a blueprint for the identification and characterization of similar ligand-binding interfaces in other IDPs regarded as "undruggable".

Small molecule inhibitors of IDPs are generally more hydrophobic and aromatic than those that target folded proteins[44]. EGCG belongs to the family of polyphenols[45] and possesses the necessary pharmacophoric properties to interact with disordered proteins. It has in fact been reported to bind and modulate the aggregation properties of IDPs such as α-synuclein[46], Aβ[47], and tau[48] by inducing the formation of structural states that prevent the formation of toxic assemblies. In addition, it was recently shown to interact with the disordered amino-terminal TAD of the tumor suppressor p53 and to induce a compact conformation, which prevented its interaction with MDM2[29]. c-MYC has been shown to be downregulated by EGCG targeting β-catenin signaling[49], though any direct association between this molecule and c-MYC has not been previously demonstrated. Here, by employing a combination of MD simulations and IM-MS, we show that EGCG interacts at multiple sites in the coreMYC peptide and drives the conformational landscape towards a more compact configuration that affects the accessibility of MBII, a major interaction site for binding partners (Fig. 5).

The finding of the apparent "specificity" in which EGCG disrupts complexes of MYC with TBP and TRRAP, but not with MAX appears surprising, considering the generally non-specific nature of EGCG binding. Even at high concentrations, EGCG engages <50% of the coreMYC population, yet completely abolishes binding to TBP and TRRAP, as judged by nMS. To better understand the basis for these observations, we considered the nature of MBII. MD analysis shows that MBII is mostly unstructured, indicating enhanced plasticity which enables a variety of binding modes that are a hallmark of fuzzy complexes. Fuzzy complex formation is a stochastic process that is controlled by the conformational landscape of disordered protein regions[50]. A prominent example is the TAD of p53, whose interactions with its folded binding partners are strongly dependent on transient structural fluctuations of a 10-residue disordered motif[51]. As shown here and elsewhere[29], EGCG modulates the conformations of disordered proteins through non-specific, often short-lived, contacts. These interactions are nevertheless sufficient to induce an imbalance in the conformational landscape of MBII that disrupts interactions with TBP and TRRAP$_{2038–2087}$. The non-specific nature of the interactions

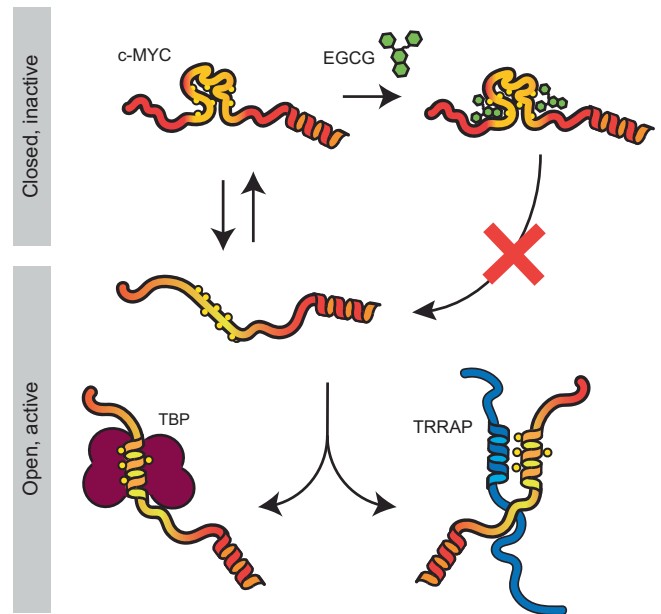

**Fig. 5 | Schematic representation of the proposed mechanism of action for coreMYC and its modulation by EGCG.** The inactive state of coreMYC adopts a compact conformation via contacts between hydrophobic residues (yellow spheres), while the active extended state with exposed hydrophobic residues enables binding to c-MYC interactors that recognizes this region of c-MYC TAD such as TBP (brown) or TRRAP (blue). EGCG stabilizes the compact state of coreMYC by interacting with the hydrophobic residues in a non-specific manner, inhibiting the formation of an open state and thus its interaction with binding partners.

implies that EGCG may also affect complex formation by changing the conformation of TRRAP$_{2038–2087}$. MAX, on the other hand, binds to the bHLHZip motif, which is more structured and therefore less likely to be affected by EGCG.

The most commonly used pharmacological intervention strategy for IDPs or proteins with intrinsically disordered regions has been to target their ordered binding partners, inhibiting their interactions based on conventional structure-based ligand design[52–54]. Alternatively, ligands that can directly interact and modulate the structure ensemble distribution of IDPs and thereby impede their interaction with endogenous binding partners is promising but largely unexplored area for targeting these challenging proteins. The identification and modulation of switch regions, such as the c-MYC TAD (as reported in this study) and p53 by EGCG are attractive developments for this strategy. In a similar manner, the small molecule SJ403 which displaces the disordered D2 domain of p27 from CDK2 has been shown to interact with D2 by causing a shift in its conformation[55]. Based on these observations, it is tempting to speculate that additional IDPs contain conformational switches that regulate access to binding sites. The major challenge with this approach, however, is the characterization of IDP-ligand interactions in fuzzy complexes, which can be dynamic with multiple binding sites. We believe that the application of probe-based MD simulations and IM-MS will provide an efficient framework for future investigations.

Finally, major efforts in the development of drugs that target c-MYC have focused on impeding the dimerization of c-MYC and MAX, either directly, or by preventing the interaction of the heterodimer with DNA[56]. A promising development in this regard has been the entry into clinical trials of the OMOMYC mini-protein[20,21], which is derived from the carboxy-terminus of c-MYC. The conformational switch identified in this study is located at the amino-terminal TAD of c-MYC, which is known to interact with multiple binding partners important for its transcriptional and oncogenic activities. In this context, we

provide a proof of principle that ligands can induce conformational changes in c-MYC, opening interesting avenues for the development of c-MYC based therapeutics. Despite the lack of specificity of EGCG binding, the insights gained from our investigation provide a potential opportunity for rational design of more specific and stronger binders to c-MYC. We envision that adding functional groups to the EGCG scaffold that specifically interact with key residues in the "pseudo-core" and increase its stability would be one potential approach. Our efforts in evoking a conformational switch hold promise for further studies aiming at identifying inducers of the inactive state of c-MYC. Our work also opens the prospects to employ similar strategies for targeting other desirable but challenging IDPs for therapeutics.

## Methods

### Modeling c-MYC peptide derivatives and system setup for MD simulations

The initial structure of all the c-MYC peptide derivatives was generated in an extended conformation using the TLEAP module of AMBER 18[57]. The amino-terminus was acetylated and the carboxy-terminus was amidated. All structures were subjected to a short MD simulation in vacuum for 200 ps and then placed in the center of a truncated octahedral box whose dimension was fixed by setting a minimum distance of 8 Å between any peptide atom and the box boundaries. The force field parameters of probe molecules (benzene, propane, methanol, and acetaldehyde) and EGCG with bcc charges was derived using the GAFF2 force-field[58] through the ANTECHAMBER module of AMBER 18. The appropriate number of probe molecules equivalent to 0.4 M concentration was computed based on the volume of the simulation box (Supplementary Table 2). The program packmol[59] was used to generate the initial distribution of the probe molecules within a sphere of radius 35 Å from the center-of-mass (COM) of the peptides. The coreMYC peptide (encompassing amino acids 101–150) system with one or two EGCG molecule was also generated with packmol by placing EGCG within a spherical distance of 15 Å from the COM of the peptide. The volume of the simulation box was kept identical for the systems with free peptides, peptides+probes, and peptide+EGCG.

### Molecular dynamics simulations

The OPC water model[60] was used for solvation and the net charge of the different peptides was neutralized by adding appropriate numbers of chloride/sodium counterions. MD simulations was carried out using the PMEMD module of AMBER 18 employing ff19SB force-field parameters[61]. All the systems were initially relaxed through energy minimization using steepest descent followed by conjugate gradient algorithms. They were then heated to 300 K over 30 ps and equilibrated for 200 ps under NVT and NPT ensembles respectively. Production dynamics was run under NPT conditions for 5 μs each for all the peptides in pure water and mixed solvent with the probe molecules, amounting to a total cumulative simulation time of 195 μs. The two systems of coreMYC with one or two EGCG molecules were also simulated for 5 μs each in duplicates. Simulation temperature of 300 K and pressure of 1 atm was maintained using Langevin dynamics[62] (collision frequency: 1.0 ps$^{-1}$) and weak-coupling[63] (relaxation time: 1 ps), respectively. Periodic Boundary Conditions (PBC) were applied on the primary simulation box and long range electrostatic interactions were computed using Particle Mesh Ewald (PME) method[64]. All the bonds involving hydrogen atoms were constrained using SHAKE algorithm[65]. The equation of motion was solved with an integration time step of 4 fs after applying the hydrogen-mass repartitioning[66]. The nine peptides ($MYC_{1-50}$ to $MYC_{401-439}$) were also subjected to MD simulations with ff14SB[67] and ff14IDPSFF[68] force-fields after solvation with TIP3P[69] water model. These peptides were simulated for 1 μs each following the same protocol and parameters as described above. The four representative coreMYC: EGCG complexes (Supplementary Figs. 9g, h and 10g, h) were used as starting structures for additional

MD simulations. These complexes were placed in the center of a truncated octahedral box whose dimensions were fixed by maintaining a minimum distance of 10 Å between any protein atom and the box boundaries. They were then solvated with OPC water model, neutralized by adding 5 Na$^+$ counter ions, energy minimized, heated and equilibrated with the same parameters as described above. The final production dynamics was done for 100 ns in five replicates for each of the four complexes.

### Modeling with AlphaFold2

The Alphafold2 (AF2) modeling in this work was done through the jupyter notebook for Colabfold (version 1.3.0)[70]. The MSA_mode was set to MMseqs2 (UniRef+Environment), pair_mode selected as "unpaired +paired", model type employed was "AlphaFold-multimer-v2" with three number of recycles, no template information was used and amber relaxation of models was disabled except in the case of coreMYC: $TRRAP_{2038-2087}$ complex. The input sequences for the modeling of c-MYC peptide derivatives in complex with PNUTS, BIN1, MAX, TBP, and WDR5 were obtained from the PDB "FASTA sequence" file of each structure. In case of generating the bound state conformation of coreMYC with the selected binding partners (Supplementary Table 1), the sequence of the isoform which has been described as canonical in the Uniprot database (Uniprot ID is shown in Supplementary Table 1) is used for model building. The sequence of the residues from 2038–2087 of TRRAP was used for predicting the peptide complex of coreMYC with $TRRAP_{2038-2087}$ and its glycine mutants (sequences shown in Supplementary Fig. 18). All the modeled complexes were ranked by multimer and the highest ranked model was selected as a representative structure for subsequent analysis.

### MD simulation of coreMYC: $TRRAP_{2038-2087}$ dimer

The highest ranked AF2 generated complex models of coreMYC with $TRRAP_{2038-2087}$ and its glycine mutant derivatives were used as starting structures for MD simulations and energy minimization respectively. These complexes were placed in the center of a truncated octahedral box whose dimensions was fixed by maintaining a minimum distance of 6 Å between any protein atom and the box boundaries. The amino- and carboxy-terminal ends were acetylated and methylated respectively, solvated with OPC water model and, the net charge of the system was neutralized by adding one Na$^+$ counter ion. The energy minimization of coreMYC: $TRRAP_{2038-2087}$ mutants and MD simulation of coreMYC: $TRRAP_{2038-2087}$ was performed using the same protocol and parameters as described above. The final production dynamics of coreMYC with $TRRAP_{2038-2087}$ was run for 1 μs in duplicate.

### Molecular modeling and MD simulation of coreMYC: TBP complex

The complex model of initial 24 amino acids of coreMYC (residues 101–124) and the core domain of human TBP (residues 159–337: uniport ID: P20226) was built guided by the structural and biochemical data reported by ref. [13]. The modeled structure was then placed in the center of a truncated octahedral box whose dimensions was fixed by maintaining a minimum distance of 6 Å between any protein atom and the box boundaries. The amino- and carboxy-terminal ends were acetylated and methylated respectively, solvated with OPC water model and the net charge of the system was neutralized by adding 8 Cl- counter ions. The simulation protocol and parameters were similar to as described above. A harmonic positional restrain of 5 kcal/mol was applied to all the backbone (N, CA, C, O) heavy atoms. The final production dynamics was run for 500 ns in duplicate.

### Computation of enthalpic and binding energies

The enthalpic energy of the seventeen c-MYC peptide derivatives and the residue-wise binding energy decomposition of coreMYC:$TRRAP_{2038-2087}$ and coreMYC:TBP structure ensemble

generated from MD simulations were computed using Molecular Mechanics/Generalized Born Surface Area (MM/GBSA) method[71] using the following equations:

$$\Delta G_{Binding} = [G]_{complex} - [G]_{peptide-A} - [G]_{peptide-B} \qquad (1)$$

$$G = E_{bon} + E_{vdw} + E_{ele} + E_{pol} + E_{npol} \qquad (2)$$

Where, $E_{bon}$ is the summation of bond, angle, and dihedral energy components, $E_{vdw}$ and $E_{ele}$ are the non-bonded Van der Waals and electrostatic interaction energies respectively. These energies are calculated using Molecular Mechanics (MM) force field expressions of AMBER. $E_{pol}$ is the polar solvation energy which is obtained by solving the Generalized Born (GB) solvation model. $E_{nonpol}$ is the non-polar solvation energy estimated through an empirical linear relationship ($\gamma$*SASA), where "$\gamma$" is the surface-tension and "SASA" is the Solvent Accessible Surface Area. For these calculations, 5000 (for the peptides) and 1000 (for the complexes) structures covering the entire simulation period were extracted, and the explicit solvent molecules were stripped. An implicit GB model (IGB = 2) was then used to model the continuum solvent environment. The dielectric constant for the solvent was set to 80, $\gamma = 0.0072$ kcal/mol/Å$^2$ and salt concentration was set to 150 mM respectively. The MMPBSA.py script[72] available through AMBER 18 was employed for computation.

### Peptide sequences of coreMYC and TRRAP$_{2038-2087}$
MGHis6-coreMYC (expressed and purified in-house)

 MGHHHHHHMVTELLGGDMVNQSFICDPDDETFI-KNIIIQDCMWSGFSAAAKLVSEKLA

 TRRAP$_{2038-2087}$ synthetic peptide (produced by GenScript, Rijswijk, Netherlands)

 Ac-GVNSVSSSIKRGLSVDSAQEVKRFRTATGAISAVFGRSQSLPGADS LLAK-NH$_2$

### Recombinant human TATA binding protein (ab81897, Abcam, Cambridge, UK)
MDQNNSLPPYAQGLASPQGAMTPGIPIFSPMMPYGTGLTPQPIQNTNS LSILEEQQRQQQQQQQQQQQQQQQQQQQQQQQQQQQQQQQQQQQQQ QQQQAVAAAAVQQSTSQQATQGTSGQAPQLFHSQTLTTAPLPGTTPL YPSPMTPMTPITPATPASESSGIVPQLQNIVSTVNLGCKLDLKTIALRARN AEYNPKRFAAVIMRIREPRTTALIFSSGKMVCTGAKSEEQSRLAARKYARV VQKLGFPAKFLDFKIQNMVGSCDVKFPIRLEGLVLTHQQFSSYEPELFPG LIYRMIKPRIVLLIFVSGKVVLTGAKVRAEIYEAFENIYPILKGFRKTT

### Expression of coreMYC
The gene encoding MGHis6-coreMYC was cloned into a pET-26b(+) vector (GenScript, Rijswijk, The Netherlands). Transformation was performed into chemically competent BL21 (DE3) *E. coli* cells (New England Biolabs, Ipswich, MA, USA). Cultures grown overnight were inoculated at 1:50 ratio into Luria-Bertani medium containing 70 mg/L kanamycin. 500 mL cultures were grown at +37 °C and induced at an OD$_{600}$ of 0.6–0.7 with 1 mM isopropyl-β-D-thiogalactopyranoside (IPTG). Expression was carried out overnight at +37 °C with continuous shaking at 150 rpm. The cultures were harvested by 20 min centrifugation at 4000 x g and the pellets were stored at −80 °C until purification.

### Purification of coreMYC
After thawing at room temperature, the pellet was resuspended in 10 mL lysis buffer consisting of 0,1 g/L lysozyme supplemented with complete mini EDTA-free protease inhibitor tablets (Roche Diagnostics GmBH, Mannheim, Germany) in Milli-Q water, followed by sonication using a probe sonicator (Branson, Brookfield, CT, USA) on ice for 10 min at 50% amplitude with pulses on/off for 5 s. The lysate

was centrifuged at +4 °C for 30 min at 12,000 x g. The supernatant was discarded and the pellet was washed with three different washing buffers, buffer 1 (1% Triton X-100, 5 mM imidazole and protease inhibitor in PBS), buffer 2 (500 mM NaCl, 1% Triton X-100, 20 mM TRIS base and 35 mM imidazole in Milli-Q water) and buffer 3 (500 mM NaCl, 1% Triton X-100, 20 mM TRIS base, 35 mM imidazole and 1,6 M urea in Milli-Q water), each time followed by centrifugation at +4 °C for 30 min at 12,000 x g. The pellet was solubilized overnight at +4 °C with continuous shaking at low speed, in a solubilization buffer containing 500 mM NaCl, 1% Triton X-100, 20 mM TRIS base, 35 mM imidazole and 7 M urea at pH 11. The solution was cleared by centrifugation for 30 min at 11,750 x g, +4 °C to remove undissolved pellet and debris. The supernatant was loaded on a His SpinTrap column (GE Healthcare, Chicago, IL, USA) equilibrated in Milli-Q water with 10 mM imidazole and 6 M urea and eluted with an elution buffer containing 250 mM imidazole and 5 M urea in Milli-Q water. The eluted fractions were evaluated on Mini-PROTEAN TGX Stain-Free 4–15% SDS-PAGE gels (BioRad, Hercules, CA, USA) and stored at −20 °C until further use.

### Native and ion mobility-mass spectrometry
MGHis6-coreMYC (hereafter referred to as coreMYC) was buffer-exchanged into 100 mM ammonium acetate, pH 7 using BioSpin6 columns (BioRad, Hercules, CA, USA). Recombinant human TBP (ab81897, Abcam, Cambridge, UK) was buffer-exchanged into 1 M ammonium acetate through double use of Zeba™ Micro Spin Desalting Columns (Thermo Fisher Scientific Inc., Waltham, MA, USA). The lyophilized TRRAP$_{2038-2087}$ peptide (GenScript, Rijswijk, Netherlands) was directly dissolved into 100 mM ammonium acetate. Concentrations for coreMYC, TRRAP peptide, and TBP were 31 µM, 480 µM, and 13 µM, respectively. The TRRAP$_{2038-2087}$ peptide stock was further diluted in ammonium acetate to a final concentration of 30 µM, to match the final concentration of coreMYC. For MS analysis, coreMYC (31 µM), TRRAP peptide (30 µM) and a mixture of both (15,5 µM coreMYC and 15 µM TRRAP$_{2038-2087}$ peptide) were prepared directly in ammonium acetate. MS analysis of coreMYC (31 mM) TBP (13 mM), or a mixture of both (6 mM coreMYC and 10 mM TBP) were prepared directly in ammonium acetate. To study the effect of EGCG, coreMYC and EGCG were mixed thoroughly by pipetting and incubated for 30 min at +37 °C. The EGCG-coreMYC solution was then added to the TRRAP$_{2038-2087}$ peptide to a final concentration of 15 mM for each peptide and 100 mM EGCG or added to TBP to a final concentration of 6 mM coreMYC, 10 mM TBP, and 100 mM EGCG. Samples were loaded into nESI capillaries (Thermo Fisher Scientific Inc., Waltham, MA, USA). Mass spectra were recorded in positive ionization mode on a Waters Synapt G1 traveling-wave IM mass spectrometer (MS Vision, Almere, The Netherlands). The capillary voltage was maintained at 1,5 kV and the sample cone was 10 V for TRRAP and 50 V for TBP measurements. The extraction cone voltage was 4 V. The source temperature was +30 °C. The trap and transfer collision energies were 10 V. The trap gas was argon at a flow rate of 4 mL/h, the IMS gas was nitrogen at a flow rate of 30 mL/h. The ion mobility-mass spectrometry settings were: IMS wave height 12 V and IMS wave velocity 350 m/s. Human insulin was used as T-Wave calibrant[73]. Data was analyzed with MassLynx version 4.1 (Waters) and Pulsar version 2.0[74]. Theoretical CCS values from pdb files were computed using IMPACT[32].

### Cell culture
Human neuroblastoma SH-SY5Y cells were cultured in 1:1 of F12 and minimal essential medium (MEM) (Cat No. 21090055, 21765037, Thermo Fisher Scientific, Waltham, MA, USA), supplemented with 10% fetal bovine serum (FBS) (Cat No. SV30160.03, Cytiva HyClone, Logan, UT, USA), 1% (v/v) L-glutamine, 1% (v/v) non-essential amino acids (Cat No. SH30034.01 and SH30238.01, Cytiva HyClone, Logan, UT, USA), and 1% (v/v) penicillin-streptomycin (Cat No. 15140122, Gibco, Life Technologies, Waltham, MA, USA) in a humidified environment at

+37 °C and 5% $CO_2$. This cell line was validated using short tandem repeat (STR) analysis (23/10/2023) (AuthentiCell, European Collection of Authenticated Cell Cultures (ECACC), UK Health Security Agency, Salisbury, UK).

## WST-1 cell viability assay

The viability of SH-SY5Y cells was analyzed using the WST-1 colorimetric assay (Cat No. 11644807001, Roche Diagnostics GmBH, Mannheim, Germany). Cells were seeded in transparent flat-bottom 96-well plate at a density of 18,000 cells per well and after 48 h the cells were treated with increasing concentrations of EGCG for a duration of 24 h. Following this period, 10 μL of WST-1 was added to each well and after one hour of incubation the absorbance was measured in a Tecan Spark 20 M multiplate reader (Männedorf, Switzerland) at 450 nm and 650 nm (as reference wavelength and subtracted during analysis) employing Tecan's SparkControl magellan version 1.2. The cell viability experiment was performed in three independent biological experiments with four replicates per repeat. Results were normalized over the 0 μM EGCG non-treated condition and statistical analysis was performed by one-way ANOVA, Dunnett´s Multiple Comparison Test, by use of GraphPad Prism version 5.0.4 (GraphPad Software Inc., San Diego, CA, USA). $p = < 0.05$ (*$p < 0.05$); $p = < 0.005$ (***$p < 0.005$). Results are displayed as mean±S.D. from $n = 3$ biologically independent experiments.

## IncuCyte live-cell imaging

SH-SY5Y cells were seeded in standard 96-well plates (TPP, Sigma-Aldrich, St. Louis, MO, USA) at a cell density of 18,000 cells per well and grown for 48 h before treatment with increasing concentrations of EGCG. Cells were then continuously imaged over a repeated schedule with image collection every two hours in the IncuCyte system (Essen Bioscience Inc., Ann Arbor, MI, USA). Analysis was performed with IncuCyte software 2021 C. For the live-cell imaging viability, $n = 3$ biologically independent experiments with four technical replicates per experimental repeat were performed. Data were processed by using GraphPad Prism version 5.0.4 (Graphpad Software Inc., San Diego, CA, USA). Results were normalized compared to starting confluence for each timepoint and condition to indicate the relative increase/decrease in confluence over time, and displayed as mean±S.D. from $n = 3$ biologically independent experiments. Statistical analysis was performed by two-way ANOVA with Bonferroni post-test. $p = < 0.0001$ (****$p < 0.0001$) indicates significance of the final data points. Results are displayed as mean±S.D. from $n = 3$ biologically independent experiments.

## In situ proximity ligation assay

SH-SY5Y cells were seeded in standard six-well tissue culture dishes (60 mm diameter) with coverslips at a cell density of 300,000 cells per well. After 48 h, cells were treated with increasing concentrations of EGCG for 24 h and fixed in 4% paraformaldehyde for 15 min at room temperature. After blocking and permeabilization (3% BSA, 0.25% Triton X-100 in PBS 1X) for 1 h at room temperature, coverslips were incubated with primary antibodies overnight in a wet chamber at +4 °C. The following antibodies were used: mouse monoclonal anti-c-MYC (1:200, 9E10, sc-40), rabbit polyclonal anti-MAX (1:200, C-17, sc-197) from Santa Cruz Biotechnology (Dallas, TX, USA), rabbit polyclonal anti-TBP (1:200, ab63766) from Abcam (Cambridge, UK), and rabbit polyclonal anti-TRRAP (1:400, SAB1300444) from Sigma-Aldrich (St. Louis, MO, USA) Following the manufacturer's protocol, secondary antibodies (anti-rabbit PLUS probe DUO92002 or anti-mouse MINUS probe DUO92004, Sigma-Aldrich, St. Louis, MO, USA) were used at 1X, and interactions detected using the Duolink® In Situ Detection Reagents Green (DUO92014). When the two proteins are at maximum distance of 40 nm, rolling circle amplification was triggered by the subsequent additions. Amplified DNA was detected by a probe containing a fluorophore ($\lambda_{ex}$ 495 nm; $\lambda_{em}$ 527 nm) and visualized using the same filter as FITC. Nuclei were counter-stained with Duolink® In Situ Mounting Medium with DAPI (Cat No. DUO82040-5ML, Sigma-Aldrich, St. Louis, MO, USA). Images were acquired with a Nikon Eclipse Ti series confocal microscope (Nikon Melville, NY, USA) equipped with a S PLAN ELWD 60X/0,70 objective (Nikon Melville, NY, USA) and a Zyla sCMOS camera (Andor, Oxford Instruments, Belfast, UK). Images were analyzed and PLA-positive signals were quantified using the NIS-Elements Advanced Research 5.02.03 software (Nikon, Melville, NY, USA). A minimum of 10 fields were captured for quantification for each condition. To analyze PLA signals, a minimum of 100 cells were randomly selected from each condition and the PLA puncta in nuclei (fixed constant area with 5 μm radius) of each cell were counted. The total number of PLA puncta was divided by the total number of nuclei for each condition to obtain an average PLA signal per nucleus. The fold decrease exhibited by EGCG treated cells were statistically compared with an ordinary one-way ANOVA, Dunnett's multiple comparison test with a single pooled variance: MYC-MAX (positive control, PC) *versus* MYC-TBP 0 μM EGCG $p = 0.0190$; MYC-MAX (PC) *versus* MYC-TBP 25 μM EGCG $p = < 0.0001$, MYC-TBP 50 μM EGCG $p = 0.0754$ (n.s), MYC-TBP 100 μM EGCG $p = < 0.0067$, or MYC negative control (NC) $p = < 0.0001$. For TRRAP statistical analysis: MYC-MAX (PC) *versus* MYC-TRRAP 0 μM EGCG $p = < 0.0001$; MYC-MAX (PC) *versus* MYC-TRRAP 25 μM EGCG $p = < 0.0001$, MYC-TRRAP 50 μM EGCG $p = 0.0038$, MYC-TRRAP 100 μM EGCG $p = < 0.0001$, or MYC negative control (NC) $p = < 0.0001$. $p = < 0.0001$ (**$p = 0.0038$; ****$p = < 0.0001$). For MYC-MAX with EGCG statistical analysis: MYC-MAX 0 μM EGCG *versus* MYC-MAX 25 μM EGCG $p = 0.0882$ (n.s.), MYC-MAX 50 μM EGCG $p = 0.9665$ (n.s.), MYC negative control (MYC NC) $p = < 0.0001$, or MAX negative control (MAX NC) $p = < 0.0001$. $p = < 0.0001$ (****$p = < 0.0001$). Data were processed by use of GraphPad Prism version 9.5.1 (Graphpad Software Inc., San Diego, CA, USA), and results are shown as mean ± S.D. from $n = 3$ biologically independent experiments. n.s= non-significant.

## Western blot

Whole-cell lysates were prepared using RIPA lysis buffer supplemented with protease and phosphatase inhibitor cocktail (Cat No. 78442, Thermo Fisher Scientific, MA, USA). Samples were boiled in 1× Laemmli buffer at + 95 °C for 5 min and separated in a Bolt™ 4–12% Bis-tris Plus gel (for MYC, MAX, TBP and β-actin) and a NuPAGE™ 3 to 8% Tris-Acetate gel (for TRRAP and α-tubulin) (Cat No. NW04120BOX and EA0375BOX, Invitrogen, Waltham, MA, USA). The PageRuler™ Plus Prestained protein ladder was used to visualize molecular weights (Cat No. 26620, Thermo Fisher Scientific, Waltham, MA, USA). Next, separated proteins were transferred into nitrocellulose membranes (Trans-Blot Turbo Transfer Pack, Cat No. 1704159, Bio-Rad, CA, USA). Membranes were blocked for 1 h in 5% non-fat milk (Cat No. A0830.5000, AppliChem GmbH, Darmstadt, Germany), and then incubated at +4 °C overnight with the following primary antibodies: goat anti-TRRAP (1:1000, sc-5405), rabbit anti-MAX mouse (1:1000, sc-197), mouse anti-β-actin (1:3000, sc-47778), and mouse anti-α-tubulin (1:2000, sc-32293) from Santa Cruz Biotechnology (Dallas, TX, USA); rabbit anti-c-MYC (1:2000, #9402) from Cell Signaling Technology (Danvers, MA, USA), and rabbit anti-TBP (1:1000, ab63766) from Abcam (Cambridge, UK). Next day, membranes were incubated with horseradish peroxidase tagged anti-mouse, anti-rabbit, or anti-goat secondary antibodies (1:3000 Cat No. P044801-2, P044701-2, or P0449, Agilent Technologies, North Billerica, MA, USA). Signals were developed using Super-Signal™ West Dura (Cat No. 34076, Thermo Fisher Scientific, MA, USA) in a ChemiDoc XRS+ System (Bio-Rad, Hercules, CA, USA). Three biologically independent replicates were performed and quantified with ImageJ software (NIH, Bethesda, MD, USA)[75]. Results were represented

as protein level relative to protein loading control (β-actin or α-tubulin) and normalized against 0 μM EGCG. Graph shows the mean ±S.D. from $n = 3$ biologically independent replicates. Statistical analysis: two-way ANOVA with * indicating $p < 0.05$.

**Real-time quantitative polymerase chain reaction (RT-qPCR)**
Total RNA was isolated using TRIzol reagent and the DirectZol RNA Miniprep Kit (Zymo Research, Irvine, CA, USA) following the instructions of the manufacturer, including DNase treatment. cDNA was synthesized using the iScript cDNA synthesis kit (Cat No. 1725037, Bio-Rad, Hercules, CA, USA). The mRNA expression was analysed with RT-qPCR using iTaq universal SYBR Green supermix (Cat No. 1725124, Bio-Rad, Hercules, CA, USA) on the StepOnePlus Real-Time PCR system (Applied Biosystems, Waltham, MA, USA). Samples were run in triplicate and the relative expression calculated using the ΔΔCT method after normalization to the mRNA levels of the internal control *β-2-microglobulin (B2M)*. Data was represented relative to the control (0 μM EGCG), as mean ±S.D. from $n = 3$ biologically independent experiments, not significant between conditions. Statistical analysis: two-tailed unpaired $t$ test. Forward (F) and reverse (R) primers used are listed in Supplementary Table 3.

**Reporting summary**
Further information on research design is available in the Nature Portfolio Reporting Summary linked to this article.

## Data availability
All data supporting the findings of this manuscript are available from the corresponding authors upon request. The molecular dynamics simulation trajectories reported in this study have been deposited in the Figshare repository (https://figshare.com/articles/journal_contribution/coreMYC/25020956). Source data are provided with this paper.

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

## Acknowledgements

The research in this study is supported by the Swedish Cancer Society grants 201288, 190480, and 190510 + 22 2266 to D.P.L., M.L. and M.A.H.; by the Swedish Research Council grants 2013–08807, 2019–01961, and 2018–02580 to D.P.L., M.L. and M.A.H.; by KI faculty grants to D.P.L., J.L.P., C.S., M.L. and M.A.H.; and by the Novo Nordisk Foundation to the NMR infrastructure facility, cOpenNMR, grant NNF18OC0032996 to K.T. C.S. is supported by a Novo Nordisk Foundation Postdoctoral Fellowship (NNF19OC0055700), J.L.P. by a postdoctoral position from the Swedish Cancer Society (22 0539), D.P.L. by a Swedish Research Council grant

for Internationally recruited Scientists (2013–08807), and M.L. by two consecutive KI faculty-funded Career Positions.

## Author contributions

DL conceived the study with input from CS, KT, DPL, ML, and MAH. DL performed all modeling analyses with computational support from LN. TV and CS purified proteins, designed and performed MS experiments, and analyzed the data. TV and JLP designed and conducted isPLA, TV and CPC performed confocal microscopy, and CPC performed data analysis. JLP performed RT-q-PCR and Western blot assays and TV cell viability experiments. DL wrote the manuscript with input from ML and MAH. All authors edited and approved the final version.

## Funding

## Competing interests

The authors declare no competing interests.
