## [Peer Review File · Nature Communications]

A druggable conformational switch in the c-MYC
transactivation domainEditorial Note: Parts of this Peer Review File have been redacted as indicated to maintain staff confidentiality.

REVIEWER COMMENTS

Reviewer #1 (Remarks to the Author):

Summary

In this work, the authors aim to present a MYC "conformational switch" from an open active conformation to a closed, inactive one. This switch was identified in a MYC fragment that the authors name "coreMYC", including the well-recognized MBII and adjacent 25-ish N-terminal residues. They induce this switch first in silico by addition of 0.4M benzene, and then by addition of a selected hydrophobic molecule, EGCG, in-vitro. Finally, they use EGCG to interrupt a MYC interaction in cellulose. Based on these observations, they propose identification of a novel avenue of SAR investigations targetting shape-shifting of MYC towards inactive conformations.

- What are the noteworthy results?

A noteworthy result could be that EGCG interrupts interactions between a MBII-containing MYC fragment and TRRAP. Extending known data on MYC-TRRAP interactions, the authors present a model that includes MYC residues not only from MBII are in contact with TRRAP, but also residues 101-110 that are outside the MBII region but overlap with the MYC TBP concave binding region as shown in Wei et al (Nature SMB 2019). The TRRAP results, however, are not substantiated by any data obtained in cellulose.

- Will the work be of significance to the field and related fields?

Ensemble modulation, conformational selection and other related strategies are already employed in this and other fields, and approaches including computational and experimental strategies have been extensively evaluated, applied and optimized (see for example Csizmek et al., Chem Rev 2016; ref 15 in this work; Heller et al., Cell Mol Life Sci 2017, and refs therein). Indeed, this body of experimental and theoretical investigations has successfully led to successful strategies for developing novel therapeutic leads (see for example Nature Rev Drug Discovery 2021, "Advances in targeting 'undruggable' transcription factors with small molecules" and refs therein).

The authors of the current work have not yet made much use of existing insights in the field. In the singular observations presented (9 fragments studied, two binding small molecules assayed in non-overlapping experiments), the narrowly selected experimental evaluation and the low level of detail in the description of the approaches taken reduce credibility of the results and even in some cases prohibits reproducibility. This significantly lowers the significance and impact of the presented work although the intention and enthusiasm by the authors and their willingness to share their preliminary results is well received.

- How does it compare to the established literature? If the work is not original, please provide relevant references.

The work presented here was stated to take a “reductionist approach” and, without any consideration of sequence, conservation or properties, divides MYC into 9 non-overlapping peptides. Notably, it has long been known that conserved regions in MYC with sequence properties different from those of IDPs form anchor points for protein interactions (first reviewed in Cowling & Cole, *Sem Cancer Biol* 2006, latest in Lourenco et al., *Nature Reviews Cancer* 2021). By the conservative 50-aa design choice of the 9 peptides, the study cuts-in-half both the MBI and the bHLZip regions, which are well-studied conserved entities and functional regions in MYC. By doing so, it is difficult to regard the work as complete since it sets the work quite apart from established literature. At the very least, overlapping peptides should have been chosen for this study, which is also normal routine in peptide screening work.

The observation that MBII is highly hydrophobic is known in the field since a long time back (reviewed in Cowling & Cole, *Sem Cancer Biol* 2006). Specifically, the MYC region comprising MYC residues 92-167, corresponding to the proposed “coreMYC” sequence in MYC (100-150), was identified as structurally protected in Fladvad et al., *JMB* 2005 as sequence-distinct from IDP regions, partly folded (“molten globule”) as judged by both CD and ANS, and structure-inducible by hydrophobic molecules (TFE). Although slightly different MYC construct and molecules were used in the submitted work, their observations regarding the properties of MBII+nterm residues has been made before and is not original.

The authors chose the Tu review (BBA, 2015) as the basis for selecting MBII-interacting proteins in an aim to investigate MYC interactions. However, this review has been superseded by the experimental and direct BioID mapping of MYC-box-dependent and -independent interactions to more than 300 proteins presented in Kalkat et al (*Mol Cell* 2018). This paper is indeed cited in the manuscript but only for its delineation of MYC-TRRAP binding. Recent structurally described but omitted MYC interactions with the MYC-TAD region include the MYC:PNUTS complex (Wei, Redel, Ahlner et al., *NAR* 2022) and the MYC:Aurora complex (Richards et al., *PNAS* 2016).

The structure of a γ TBP-hMYC complex was recently determined by a combination of crystallography, NMR and modelling (*Nature SMB*, 2019). This structure was directly used in this work to derive a model for hTBP with part of the “coreMYC” sequence. Since hTBP and γ TBP are highly conserved, doing such a model based on existing high-resolution data is very straight-forward and not a main, original result.

The authors do not critically evaluate the structural and functional differences between the MYC-TRRAP complex studied here with that shown in Feris et al., 2019 and originally identified in McMahon et al (*Cell* 1998). This would have added novelty to this work and would have extended current knowledge.

- Does the work support the conclusions and claims, or is additional evidence needed?
- Are there any flaws in the data analysis, interpretation and conclusions?

The authors state that MYC cycles between closed, inactive and open, active conformations. To support this statement, the authors should make use of their extensive MD simulations and show that the conformations are actually in equilibrium (“cycles” or “fluctuates”) by in-depth analysis of the MD trajectories as a function of time.

As mentioned above, the credibility of the work is significantly limited by the choice of non-overlapping MYC peptides, and also by the limited choice of interacting compounds. To be credible, the authors should extend their MYC peptide library to include overlapping peptides, taking into consideration already well-known MYC binding epitopes, and increase the chemical scope and range of interacting compounds assayed.

It is well known that MD force fields that well reproduce properties of folded proteins do not always reproduce IDP properties accurately. Due to the high reliance on in-silico methods in this work, the authors should carefully motivate their use of forcefield, water model and sampling methods as well as ensure reproducibility to experimental data by a direct method to determine R_g , such as SAXS, for at least a subset of the peptides.

The mixed-solvent MD approach is commonly used for fragment screening of folded proteins to identify hydrophobic pockets, but to my knowledge this approach has not been benchmarked as an established procedure for IDPs. Therefore, it is not evident whether the high (0.4M!) concentration of benzene induces solvent co-aggregation in silico (actual aggregation is not possible due to the positioning of each MYC peptide in a different solvated box), whether it by time restrains the conformational sampling to a single state, or whether the benzene bound or -free conformations are in equilibrium. Also here, careful evaluation and presentation of the MD simulations as a function of time is of essence.

Row 182: Discussion on the enthalpic energy as favorable or unfavorable is misleading since the entropic contribution, which is a very significant contribution to the free energy in the extended conformations, is entirely ignored. Since the authors only rank the states based on the enthalpic energy and entirely dismiss entropy, the energy scores are biased towards states bound to as many benzene molecules as possible, which is also presented in figure 1D. This ranking prevents the authors to rank the presence of “disorder-to-more-disorder” transitions, where extensive entropy contributes favorably to the free energy. The conclusion that the “high-energy extended states of coreMYC with exposed hydrophobic residues” is thus entirely misled by the chosen energetic ranking (enthalpy-based) and is therefore not supported by the data presented.

Conclusions from the experimental native mass spectrometry evaluation of EGCG binding to MYC appear biased towards the EGCG-bound state, which, according to Fig 2b, is only present in small amounts and at high excess of EGCG. Furthermore, the arrival time distribution is normalized and presented in an overlay with the unbound states, where the propensity of the longer arrival times can barely be discerned as being put behind the unbound sample, thereby graphically exaggerating the propensity of the shorter arrival time. Notably, even with 2 EGCG molecules bound, the distribution rather appears to be 50-50 between closed (short arrival times) and “open” (longer arrival times), rather than a near-complete conversion suggested from MD simulations as emphasized in conclusions made by the authors. Given the relatively small signal for the EGCG-bound states, it would be relevant to show the errors in the distribution intensity for the 0, 1 and 2 EGCG molecules bound. The situation is similar for the data presented in figure 3c: only a very small portion of the TBP proteins are found to be in complex with MYC, thus weakening the conclusions that can be drawn from this experiments.

The AF modeling cases (row 323 and onwards) are presented with no reporting of scores, in particular

the iptm is lacking, which is the score for the interaction. Without these scores, it is impossible to judge the quality of the predicted structures (presented in Fig 3a,b, Fig S5). Furthermore, the MYC peptide docking to these structures in AlphaFold (using ref 31) is not based on the most recent multimer version of AlphaFold but instead uses the glycine linker “trick” to support formation of the complex in-silico, thereby limiting the number of attainable complex models and thus the conclusions that can be drawn from these docking experiments. For an up-to-date and accurate benchmark on how to use peptide-docking for multimer this is the proper reference: 10.3389/fbinf.2022.959160, Johansson et al. Finally, with the exception of the modelling of the coreMYC:TRRAP and coreMYC:TBP complexes, the manuscript has no methods section pertaining to the AF modeling part of the work, nor any description of the modelled complexes presented in Suppl Fig 5, their similarity to already determined structures of the target proteins, or if they were modelled in their entity. Taken together, this lack of evaluation info and inappropriate approaches severely limit this section of the work as presented in its current state.

In cellulo, the authors only show interruption of the coreMYC:TBP interaction, not the interruption of coreMYC:TRRAP. Given that the work presents analysis of these two interactions both by MD, AF and ESI, the coreMYC:TRRAP interaction should also be probed in cells. Furthermore, it was shown in Wei et al (Nature SMB 2019) that the interaction critical for tumor formation is not hydrophobic, but electrostatic, and focused on the convex side of the MYC:TBP complex instead of the amphiphatic helix bound to the concave, inner side of TBP. This renders the MYC:TBP interaction a peculiar biological control in the light of MYC inhibition model and therapeutic strategy suggested by the authors.

In Figure 4, the concentration of EGCG used in the MYC:TBP PLA experiment (25 μ M) exceeds the concentration where 20% of the cells are already showing reduced viability (Fig S10). Furthermore, the control experiment made to show dose-dependent reduction of nuclear MYC-TBP complexes is performed at even higher EGCG concentrations (50 and 100 μ M), at which the viability of the cells is decreased to below 50%, which surely must affect the outcome of the experiment. Furthermore, the obvious control experiment treating the MYC-MAX complex similarly with EGCG as the MYC-TBP complex to investigate specificity is to my understanding not done or not shown. The apparent presence of significant amounts of TBP in the cytosol is bewildering. Together, this adds unreliability to the cellular conclusions in this work.

Short comments:

Ref 27 should rather be Ref 28.

Figure S1 should be a line plot (as in Fig 1c) since curves behind the front data display are concealed and thereby cannot be evaluated or discussed.

- Do these prohibit publication or require revision

As of now, and as described above, this work has many features of a preliminary report and will need to be significantly strengthened in methodological, experimental and strategic aspects as well as in in-depth understanding of IDPs and their interactions. Current methodological progress in the IDP field needs to be considered, as well as reflection on concepts such as folding-on-binding versus conformational

selection and entropy as a driving force for interaction. Furthermore, the study is still limited by the small number of small molecules assayed (only two) and the limited number of methods used.

- Is the methodology sound? Does the work meet the expected standards in your field?

The IDP field has developed with incredible speed during the past couple of years, with lots of experimental approaches tailored to the analysis of IDP interactions. For a complex system such as this and given the extensive in-silico part of the investigation, at least one more biophysical method would be required to currently meet the expected standards in the IDP field. Normally this would be NMR, but it could also be a fluorescence method if solubility and/or access of the peptides/proteins is low. Note that no experiments in this work directly address affinity which leaves a methodological void.

- Is there enough detail provided in the methods for the work to be reproduced?

As described in my detailed review above, in several instances detail or even entire sections in methods are lacking, which in several instances precludes reproducibility.

Reviewer #2 (Remarks to the Author):

High quality study described in a well crafted manuscript.

In this study, the authors used molecular dynamic (MD) simulations and mass spectrometry to identify a conformational switch region of MYC, termed coreMYC.

Indeed, MD simulations initially enable them to identify a core region of the intrinsically disordered MYC protein that displays a different degree of compactness in presence of water or benzene molecules. They then proceed to characterise the conformational landscape of this region using recombinantly produced 50 aa peptide and native mass spectrometry, in absence and presence of EGCG and to model the bound complexes of coreMYC to known MYC interactor proteins. Their results suggest that the more high-energy extended states of coreMYC constitutes an active recognition module, while the compact conformation represents an inactive state of this peptide portion of MYC. Using in situ PLA, they observe a reduction of c-MYC:TBP complexes in neuroblastoma cells in presence of EGCG.

Findings are novel and the work opens significant and exciting avenues for drugging the IDP region of MYC.

The methodology is sound, and the work supports the conclusions and claims.

Minor comment:

Does EGCG influences (reduces) MYC or TBP protein levels in SH-SY5Y cells? This appears like an important control to be able to conclude on the impact of EGCG on reducing the MYC:TBP complexes observed by isPLA: is the reduction of those complexes upon EGCG treatment due to the adoption of an inactive conformation of the switch or simply to a reduction of MYC protein levels?

The authors should include a WB control for MYC protein levels before and after treatment with EGCG and ideally a qPCR assessing possible changes in MYC mRNA transcript.

Reviewer #3 (Remarks to the Author):

The manuscript by Lama and coworkers describes probing conformational landscape of human c-MYC transcription factor using cosolvent (probe-based) molecular dynamics (MD) simulations and protein modelling, with validation provided by native mass spectrometry (MS) and cell-based assays. Using this approach, the authors identified a conformational switch within the N-terminal transactivation domain of c-MYC, which has not been reported previously. This switch shifts between a closed (inactive), and an open (active) conformation of the protein. Furthermore, the authors showed how the conformational landscape of c-MYC can be shifted towards its inactive conformation by non-specific small molecule binding.

This is highly significant, since c-MYC is the most frequently deregulated oncogene across the majority of human cancers. As most of its sequence lacks of a stable tertiary structure, c-MYC has been deemed “undruggable” by conventional structure-guided drug discovery approaches. Although the idea of shifting the ensemble of intrinsically disordered proteins is not new (see e.g. Kriwacki et al., J. Am. Chem. Soc. 2017, 139, 39, 13692), any approach that tackles the “druggability” of c-MYC deserves attention. Induction of closed conformation impedes interactions of c-MYC with the endogenous binding partners such as TRRAP and TBP (TATA-box binding protein), which are crucial for the oncogenic activity of c-MYC. Small molecule inducers of closed conformation may therefore represent an alternative strategy for “drugging” c-MYC. While the results presented in the manuscript are very early-stage and unlikely to translate into therapeutics any time soon, they represent a valid proof-of-concept.

The manuscript is very well-written, supported by high quality figures and supplementary information, and I thoroughly enjoyed reading it. The work is novel, and the outcomes would be of interest to the medicinal chemistry and cancer research communities. I have some questions regarding the methodology of the simulations, and I would like to suggest changes which would improve the manuscript. I would suggest acceptance after addressing these questions and comments.

1. It should be clarified why 50 residues segments were selected for independent simulations: the number seems somewhat arbitrary.
2. While most of c-MYC is intrinsically disordered, some of its segments show elements of secondary/tertiary structure, which were captured by structural biology approaches (e.g. bHLH). As I understand, all 50-residues segments were modelled in fully extended polypeptide conformations and subjected to MD simulations. While radii of gyration plots (Fig. 1) show various degrees of compactness acquired during the simulation, there is no comparison between the most-populated clusters obtained during MD and those few segments that were solved by X-ray crystallography and NMR (e.g. PDB codes 2A93, 5I4Z). It would be helpful to show the secondary structure plots (e.g. DSSP) for those regions.
3. Cosolvent (probe-based) MD simulations: benzene (0.4 M), which has been used as a cosolvent/probe, has been noted to undergo phase separation due to stacking interactions in extended MD simulations in aqueous solvents. Production trajectories reported in the manuscript were 5 microseconds. Was any of the phase separation of the hydrophobic probe observed? There is no information about it, and I am curious how the phase separation have been avoided.

4. Although ff19SB and OPC are a good choice for the force field and solvent models for IDPs, a comment in the discussion on the selection/choice of the most appropriate force field should be added.

5. MD simulations: all production runs were performed in duplicates. How consistent were the observables obtained in each replica, e.g. radii of gyration? Were the results showed in Fig. 2b the results of averaging, or the trajectories were concatenated/combined?

6. My concern is that only one type of cosolvent (benzene) and only one type of molecular probe (EGCG) were used in the study. The results would be a lot more compelling if several different cosolvents (hydrophobic, hydrophilic, aromatic, aliphatic, etc) were employed. Likewise, IDP modulators other than EGCG (see e.g. Berkeley and Debelouchina, Chem Sci 2022 Nov 21;13(48):14226) should be considered. EGCG is fine as a tool compound for IDP modulation *in silico* and *in vitro*, but because of its low affinity, low selectivity, toxicity in concentrations over 20 micromolar (mentioned in the manuscript), and poor bioavailability, it is unlikely to be used as a lead. It might be that better suited alternatives for ensemble-shifting exist.

7. EGCG binding modes: although it is likely that EGCG exerts its effects via non-specific interactions, it would be recommended to further assess the predicted binding modes by running several shorter simulations (50 – 100 ns) with different starting coordinates of EGCG and monitoring the convergence of the binding poses showed in Fig. 2. An addition of molecular docking (using the closed conformation of coreMYC segment as a target) would also be beneficial in characterisation of those interactions.

Reviewer #4 (Remarks to the Author):

In this manuscript Lama et al describe the use of strategy combining computer simulations native mass spectrometry and cell biology to understand the conformational state(s) adopted by the intrinsically disordered protein c-Myc. They authors show how these conformations can be modulated within a precise region of c-Myc by small molecules and that this has consequences on the ability of c-Myc to participate in protein interactions. This is an elegantly designed and conducted study and the manuscript describes the data very well indeed. Here I highlight a few minor areas that the authors may wish to comment on when revising their manuscript for publication, focusing on the mass spectrometry elements of the work.

- Line 245 – The authors state that Rg is analogous to CCS. They should, however, cite some literature here discussing possibilities for protein compaction and other gas phase structural rearrangement phenomena that must be considered when interpreting ion mobility data.

- Line 250 – The authors describe that the data show three major populations. However, on close inspection it looks as if the multimodal arrival time distribution could be fitted to at least 4 conformers (represented by Gaussian distributions). Are the authors calling the species between 2-4 ms one compact state? If so, the authors should consider adding labels to Figure 4c that splits the ATDs into

these three states.

- Are similar structural effects observed for the 5+ charge state of coreMYC? Even if not, the authors should report CCS values for this species and show that structural changes can or cannot be detected upon EGCG binding.
- Line 269. How were the compact states of coreMYC selected from the MD simulations and what method/software was used to calculate the theoretical CCS? Are the experimental CCSs calculated from the peak top arrival time values? The authors should also check that their reporting of the ion mobility data are compliant with recommendations set out in PMID: 30707468.
- Have the authors checked to see if EGCG binding modulates the conformations of full length c-Myc by ion mobility MS?
- Line 356. The authors conclude that EGCG inhibits the ability of coreMYC to interact with TRRAP. Does EGCG bind to TRRAP? How do the authors know that the inhibition is due to binding Myc and not TRRAP? A control experiment should be performed to check this and the language amended accordingly.
- Figure 3c. Why is only one charge state detected for the coreMYC+TRRAP complex? Could the authors please comment on this.

Reviewer #1 (Remarks to the Author)

Summary

In this work, the authors aim to present a MYC "conformational switch" from an open active conformation to a closed, inactive one. This switch was identified in a MYC fragment that the authors name "coreMYC", including the well-recognized MBII and adjacent 25-ish N-terminal residues. They induce this switch first in silico by addition of 0.4M benzene, and then by addition of a selected hydrophobic molecule, EGCG, in-vitro. Finally, they use EGCG to interrupt a MYC interaction in cellulose. Based on these observations, they propose identification of a novel avenue of SAR investigations targeting shape-shifting of MYC towards inactive conformations.

- What are the noteworthy results?

A noteworthy result could be that EGCG interrupts interactions between a MBII-containing MYC fragment and TRRAP. Extending known data on MYC-TRRAP interactions, the authors present a model that includes MYC residues not only from MBII are in contact with TRRAP, but also residues 101-110 that are outside the MBII region but overlap with the MYC TBP concave binding region as shown in Wei et al (Nature SMB 2019). The TRRAP results, however, are not substantiated by any data obtained in cellulose.

We fully agree with the referee that it is important to perform experiments to analyze the effect of EGCG on the MYC-TRRAP interaction in cells. Thus, we have now included isPLA data showing that EGCG disrupts the MYC:TRRAP interaction in cellulose, which confirms the mass spectrometry and AlphaFold data. Please see Figures 4 and S22.

- Will the work be of significance to the field and related fields?

Ensemble modulation, conformational selection and other related strategies are already employed in this and other fields, and approaches including computational and experimental strategies have been extensively evaluated, applied and optimized (see for example Csizmek et al., Chem Rev 2016; ref 15 in this work; Heller et al., Cell Mol Life Sci 2017, and refs therein). Indeed, this body of experimental and theoretical investigations has successfully led to successful strategies for developing novel therapeutic leads (see for example Nature Rev Drug Discovery 2021, "Advances in targeting 'undruggable' transcription factors with small molecules" and refs therein).

The authors of the current work have not yet made much use of existing insights in the field. In the singular observations presented (9 fragments studied, two binding small molecules assayed in non-overlapping experiments), the narrowly selected experimental evaluation and the low level of detail in the description of the approaches

taken reduce credibility of the results and even in some cases prohibits reproducibility. This significantly lowers the significance and impact of the presented work although the intention and enthusiasm by the authors and their willingness to share their preliminary results is well received.

We are grateful for the reviewer's appreciation. Although we do not consider the complementary insights from MD simulations, native mass spectrometry, and in cellulo assays singular observations, we agree that further experimental support could strengthen the conclusions. As outlined below, we have therefore (a) reinforced the MD results with eight overlapping peptides and three additional molecular probes, (b) included additional MS and tryptophan fluorescence data to extract relative binding affinities of EGCG for coreMYC, (c) confirmed both EGCG binding to MYC box II and the resulting compaction of coreMYC by NMR spectroscopy, (d) confirmed the effect of EGCG on the TRRAP interaction with isPLA data, and (e) showed that EGCG does not affect protein levels of MYC, MAX, TRRAP, or TBP nor *c-MYC* mRNA expression. Please see further explanations of these experiments below.

- How does it compare to the established literature? If the work is not original, please provide relevant references.

The work presented here was stated to take a "reductionist approach" and, without any consideration of sequence, conservation or properties, divides MYC into 9 non-overlapping peptides. Notably, it has long been known that conserved regions in MYC with sequence properties different from those of IDPs form anchor points for protein interactions (first reviewed in Cowling & Cole, *Semin Cancer Biol* 2006, latest in Lourenco et al., *Nature Reviews Cancer* 2021). By the conservative 50-aa design choice of the 9 peptides, the study cuts-in-half both the MBI and the bHLHzip regions, which are well-studied conserved entities and functional regions in MYC. By doing so, it is difficult to regard the work as complete since it sets the work quite apart from established literature. At the very least, overlapping peptides should have been chosen for this study, which is also normal routine in peptide screening work.

The primary objective with computational peptide design of *c-MYC* was to screen for an epitope within the intrinsically disordered protein whose structural ensemble can be modulated with exogenous ligands. We designed nine non-overlapping *c-MYC* peptides in an unbiased manner that led to the identification of a peptide derivative (residues 101-150; termed coreMYC) whose conformational distribution was particularly sensitive to the presence of benzene probes. This epitope was then targeted with a small molecule (EGCG) that inhibited protein-protein interaction between *c-MYC* and two of its most well-characterized and central binding partners, TBP and TRRAP, both with importance for the tumorigenic activity of *c-MYC*. Thus, the set of designed peptides was adequate in enabling us to successfully identify an epitope with a potential druggable feature against *c-MYC*.

However, taking note of the reviewers comment, we have now designed an additional set of eight 50-residue peptide derivatives (Pep_10 to Pep_17) overlapping with the initial set of nine peptides (Pep_1 to Pep_9; **Figure Aa**), and subjected them to MD simulations with and without benzene. The resulting R_g frequency distribution profiles of Pep_10 to Pep_17 do not show any considerable variation between the two solvent conditions, with a relative transition towards a more compact state observed for some peptides in the presence of benzene (**Figure Ab**). A comparison across all seventeen derivatives (non-overlapping and overlapping) highlight that the conformations of the peptides are influenced to different degrees by the addition of benzene, but the impact on the ensemble distribution is most substantial (R_g distribution range from 5-35 Å in water to a narrow range between 5-15 Å in water+benzene) for coreMYC. Collectively, the probe-based peptide screening with overlapping peptides further underscores the distinct sequence composition of the coreMYC epitope, which makes it particularly susceptible to conformational modulation by benzene. We have now added this result in the revised version of the manuscript (page 6, Figure S3).

Figure A. Design and conformational sampling of c-MYC peptide derivatives. (a-b) Representation of the seventeen peptide derivatives of c-MYC analyzed. The nomenclature and the first residue number (as well as the last for Pep_10 and Pep_17) of each derivative are indicated. **(c)** Normalized frequency distribution of R_g computed from the ensemble of structures generated from explicit solvent MD simulations in water (black) and water + benzene (red) for the eight (Pep_10 to Pep_17) c-MYC derived peptides whose residue range are indicated.

The observation that MBII is highly hydrophobic is known in the field since a long time back (reviewed in Cowling & Cole, *Sem Cancer Biol* 2006). Specifically, the MYC region comprising MYC residues 92-167, corresponding to the proposed “coreMYC” sequence in MYC (100-150), was identified as structurally protected in Fladvad et al., *JMB* 2005 as sequence-distinct from IDP regions, partly folded (“molten globule”) as judged by both CD and ANS, and structure-inducible by hydrophobic molecules (TFE). Although slightly different MYC construct and molecules were used in the submitted work, their observations regarding the properties of MBII+nterm residues has been made before and is not original.

We thank the reviewer for highlighting these references. It is interesting that the druggable epitope (residues 101-150; termed coreMYC) identified in our study is a region that resides within the TAD (residues 92-167) which was previously defined as structurally protected (Fladvad et al., *JMB*, 2005). However, besides detecting and highlighting the distinct physico-chemical property of coreMYC, we have more importantly shown that the structural ensemble of this epitope can undergo a ligand-induced conformational switch. We have exhibited the plasticity transitions between extended and compact states of coreMYC, and described the functional implication of this flexibility. Our finding shows that the extended states with the exposed hydrophobic residues facilitates interaction with a range of protein co-factors, whereas the compact states driven by hydrophobic collapse shield these residues from the aqueous environment thereby preventing protein recognition. We have further demonstrated with the use of EGCG that coreMYC can be induced to adopt a more compact conformation that prevents its binding to TRRAP and TBP both in vitro as well as in cells. Thus, the originality of our work stems not only from the identification of a cryptic epitope within the intrinsically disordered MYC protein, but from the functional characterization of the conformational states associated with this epitope and the potential to target this region with shape-shifting compounds for therapeutics. We have included the study by Fladvad et al., (*JMB*, 2005) in our discussion of the revised manuscript (page 23).

The authors chose the Tu review (BBA, 2015) as the basis for selecting MBII-interacting proteins in an aim to investigate MYC interactions. However, this review has been superseded by the experimental and direct BioID mapping of MYC-box-dependent and -independent interactions to more than 300 proteins presented in Kalkat et al (*Mol Cell* 2018). This paper is indeed cited in the manuscript but only for its delineation of MYC-TRRAP binding. Recent structurally described but omitted MYC interactions with the MYC-TAD region include the MYC:PNUTS complex (Wei, Redel, Ahlner et al., *NAR* 2022) and the MYC:Aurora complex (Richards et al., *PNAS* 2016).

The coreMYC epitope (residues 101-150) harbours the functionally important MBII motif (residues 128-143) which serves as a binding site for multiple c-MYC protein

interactors. We thus asked whether the observed conformational transitions of coreMYC between compact and extended states could regulate MBII recognition of its binding partners. To address this query, we created a dataset of c-MYC interactors by carefully selecting eighteen different proteins, which has been characterized to physically interact with c-MYC via MBII and/or the amino-terminal region including coreMYC (Tu et al., BBA, 2015). Alphafold2 (AF2) generated models of coreMYC when bound to each of the eighteen interactors indicated that the epitope adopts a predominantly extended conformation in these complexes, underlying that the extended state is potentially the active recognition module. We can further expand the data set to include additional MBII dependent c-MYC interactors (Kalkat et al., Molecular Cell, 2018). However, we believe that the observations drawn from this part of the study are reasonable to postulate that the extended conformation represents a functionally important state for protein recognition by coreMYC. Thus, this realizes our aim to evaluate the potential active conformation of coreMYC, which was validated experimentally by inhibition of its interaction with TRRAP and TBP (Figure 3). We thank the reviewer for bringing to our notice the structural characterization of PNUTS and Aurora with the TAD of c-MYC and MYCN, respectively. We have now included both these studies in the Introduction section of our revised manuscript (page 3).

The structure of a yTBP-hMYC complex was recently determined by a combination of crystallography, NMR and modelling (Nature SMB, 2019). This structure was directly used in this work to derive a model for hTBP with part of the "coreMYC" sequence. Since hTBP and yTBP are highly conserved, doing such a model based on existing high-resolution data is very straight-forward and not a main, original result.

The core domain of hTBP and yTBP indeed have a high degree of sequence identity (~81%), and we agree with the reviewer that template-based modelling under this condition is a standard procedure with a high level of confidence. However, we have not claimed that showing this interaction is a novel finding. Instead, we first demonstrate dimerization between full length hTBP and coreMYC using native MS. Subsequently, we show that pre-incubation of coreMYC with EGCG completely abolishes this complex formation confirming direct impact of EGCG on the active conformation of coreMYC. Hence, our main/original finding is that the highly important interaction identified by Sunnerhagen and co-workers can be successfully targeted with a small molecule. Our modelling data indicates that coreMYC recognizes hTBP in an extended conformation involving hydrophobic interactions, which rationalizes that treatment of coreMYC with EGCG drives the peptide towards a hydrophobic collapsed state (another main result), and thereby abrogates its interaction with hTBP. Thus, the observation from the standard template-based modelling here is a complementary molecular description to the novel observed inhibition of coreMYC: hTBP interaction by EGCG.

The authors do not critically evaluate the structural and functional differences between the MYC-TRRAP complex studied here with that shown in Feris et al., 2019 and originally identified in McMahon et al (Cell 1998). This would have added novelty to this work and would have extended current knowledge.

In this work, we have carried out native MS of c-MYC₁₀₁₋₁₅₀ (coreMYC) and TRRAP₂₀₃₈₋₂₀₈₇, which form a 1:1 complex suggesting specific association. This finding underscores through an orthogonal method that TRRAP is an MBII (residues 128-143) dependent co-factor of c-MYC, which as the reviewer has rightly points out was previously identified and demonstrated (McMahon et al., Cell, 1998; Feris et al., PLoS One, 2019). However in addition and more importantly, we have shown using native MS that a small molecule (EGCG) can inhibit the interaction between coreMYC and TRRAP₂₀₃₈₋₂₀₈₇. Further, through a combination of AF2 and MD simulations, we have developed a bound state model of the peptides. Feris et al., (PLoS One 2109) have shown that the interacting peptides from c-MYC and TRRAP undergo a "disorder-to-order" transition in the formation of a structurally-stable conformation with high alpha helical character; and that residue W135 within the MBII motif is directly involved in the recognition of TRRAP. Our structural model suggests an elongated and predominantly helical structure of the two peptides in the complex state (**Figures 3 and S18**). We also find that W135 is an integral part of the hydrophobic interaction network between coreMYC and TRRAP₂₀₃₈₋₂₀₈₇. Thus, these correlations provide good support for the proposed nature of the interaction between the two peptides. Based on this model, we have also derived in silico glycine mutants of W135 as well as other hydrophobic residues in the MBII motif. The binding energies of the AF2 generated complexes of coreMYC mutants with TRRAP₂₀₃₈₋₂₀₈₇ indicate that the potency of interaction decreases with the replacement of hydrophobic residues (**Figure S19**). Collectively, we believe that our MS, AF2, and MD data on coreMYC:TRRAP₂₀₃₈₋₂₀₈₇ complexation extends the knowledge by providing critical insights into their potential mode of interaction, the ability to disrupt this interaction with a small molecule, and the specific residues in the MBII motif that are involved in the recognition.

- Does the work support the conclusions and claims, or is additional evidence needed?
- Are there any flaws in the data analysis, interpretation and conclusions?

The authors state that MYC cycles between closed, inactive and open, active conformations. To support this statement, the authors should make use of their extensive MD simulations and show that the conformations are actually in equilibrium ("cycles" or "fluctuates") by in-depth analysis of the MD trajectories as a function of time.

As suggested by the referee, we have now included a R_g plot of coreMYC as a function of the simulation time with representative compact and extended state conformations of the peptide in the revised version of the manuscript (**Figure S12b**).

As mentioned above, the credibility of the work is significantly limited by the choice of non-overlapping MYC peptides, and also by the limited choice of interacting compounds. To be credible, the authors should extend their MYC peptide library to include overlapping peptides, taking into consideration already well-known MYC binding epitopes, and increase the chemical scope and range of interacting compounds assayed.

We have now extended the peptide screening of c-MYC with an additional set of eight 50-residue derivatives (Pep_10 to Pep_17; **Figure Aa**) overlapping with the initial set of nine peptides (Pep_1 to Pep_9; **Figure Aa**), and subjected them to MD simulations with and without benzene. The resulting R_g frequency distribution profiles indicate that the conformations of the peptides are influenced to different degrees by benzene with transition towards a more compact state especially for Pep_11, which is a derivative that overlaps with coreMYC. However, none of these additional peptides undergo any significant ensemble modulation between the two solvent conditions as observed for coreMYC (**Figure Ab**).

In addition, we have now also subjected coreMYC to additional mixed-solvent simulations in the presence of three different molecular probes, which include propane (hydrophobic: aliphatic), methanol (hydrophilic: h-bond donor), and acetaldehyde (hydrophilic: h-bond acceptor) (**Figure B**). We find that propane shifts the structural ensemble of coreMYC towards a compact conformation as observed for benzene (hydrophobic: aromatic), while the hydrophilic probes (methanol and acetaldehyde) marginally increases the population of the compact state, but have no significant impact on the epitope. Propane induces the conformational transition by acting as a molecular "glue" (like benzene) in stabilizing the hydrophobic collapsed state of coreMYC. These mixed-solvent simulations with multiple and diverse chemical fragments further highlights the distinct sequence composition of coreMYC which makes it particularly susceptible to conformational modulation by certain molecular probes. It also supports the potential of fragment-based ligand design and development with specific pharmacophoric properties to target coreMYC as exemplified by the use of EGCG (hydrophobic, aromatic, and h-bond donor). We have now added this result in the revised manuscript (page 7; **Figure S7**).

Figure B. Mixed-solvent simulations of coreMYC. (a-c) R_g -E (energy normalized in the scale from 0 to 1) distribution plots of coreMYC in mixed-solvent simulations of water with (a) propane (green), (b) methanol (orange), and (c) acetaldehyde (blue). R_g -E distribution of coreMYC in water (black) is shown for reference in all three plots. (d) Normalized frequency distribution of R_g computed from the ensemble of structures generated from explicit solvent simulations in pure water and mixed-solvent simulations of water with propane, methanol, or acetaldehyde.

It is well known that MD force fields that well reproduce properties of folded proteins do not always reproduce IDP properties accurately. Due to the high reliance on in-silico methods in this work, the authors should carefully motivate their use of forcefield, water model and sampling methods as well as ensure reproducibility to experimental data by a direct method to determine R_g , such as SAXS, for at least a subset of the peptides.

We are aware that one of the major concern with atomistic simulations of IDPs employing currently available protein force-field is their tendency to favour overly compact structures (Huang et al., *Curr Opin Struct Biol*, 2018; Shabane et al., *JCTC*, 2019). In this regard, we have indeed carried out an independent assessment of three different AMBER-based protein force-field (ff14SB, ff14IDPSFF, and ff19SB) with two water models (TIP3P and OPC) to sample the conformational states of all the initially designed nine non-overlapping c-MYC peptides (Figure C). We found that "ff14SB + TIP3P" (Maier et al., *JCTC*, 2015), which is one of the standard combination of force-field and water model used in AMBER derived simulations sampled only the collapsed

states for all the nine peptides. The combination of ff14IDPSFF (Song et al., JCI, 2017; a force-field tailored for IDPs) and TIP3P relatively improved the sampling property of the peptides. Significantly, the most recently developed AMBER force-field ff19SB (Tian et al., JCTC, 2020), which has been parametrized with OPC water model was able to emulate the highest structural diversity with exploration of extended states across the different peptides. Based on this evaluation, we have performed our simulations with ff19SB force-field and OPC water model. We have now included this section (as Supplementary text) and Figure (Figure S1) in the revised manuscript.

We agree that a direct benchmarking of the measured radius of gyration from simulations with experimentally derived value for the peptides would be ideal in order to accurately evaluate the sampling efficiency of our simulations. However, the specific purpose of the simulations was to compare the conformational sampling of the peptide derivatives in the presence and absence of molecular probes for epitope identification. On that account, it was rewarding to note that the conformation of only one peptide (residue 101-150; coreMYC) was significantly influenced by shifting towards a compact state with the addition of benzene. Further, this MD observed ensemble modulation of coreMYC was validated with native MS and IMMS using a polyphenolic compound, EGCG, providing experimental support for the effectiveness of the designed MD methodology in mapping ligand inducible shape-shifting epitopes.

Figure C. Assessment of force-fields and water models. Normalized frequency distribution of R_g computed from the ensemble of structures generated from 1 μ s explicit solvent MD simulations for the library of nine non-overlapping c-MYC peptides in three different combinations (ff14SB+TIP3P, ff14IDPSFF+TIP3P, and ff19SB+OPC) of AMBER-based protein force-fields and water models. The peptides and the residue range are indicated.

The mixed-solvent MD approach is commonly used for fragment screening of folded proteins to identify hydrophobic pockets, but to my knowledge this approach has not been benchmarked as an established procedure for IDPs. Therefore, it is not evident whether the high (0.4M!) concentration of benzene induces solvent co-aggregation in silico (actual aggregation is not possible due to the positioning of each MYC peptide in a different solvated box), whether it by time restrains the conformational sampling to a single state, or whether the benzene bound or -free conformations are in equilibrium. Also here, careful evaluation and presentation of the MD simulations as a function of time is of essence.

Mixed-solvent simulations has indeed traditionally and successfully been used to reveal cryptic potential ligand-binding sites on surfaces of well-folded protein domains (Ghanakota et al., JMC, 2016; Kimura et al., JCI, 2017). In this study, we have extended its application to disordered peptide derivatives from c-MYC as a means to screen for an epitope within the protein whose structural ensemble can be modulated with exogenous ligand. We concur with the reviewer that this could indeed be the first such application of mixed-solvent simulations of an IDP, which can have wide-ranging implications in future identification and characterization of ligand-binding interfaces on disordered proteins.

We have carefully examined for the co-aggregation of benzene probes in our simulations, and at a concentration of 0.4M, the molecules were soluble and we did not find any significant phase separation. In fact we have previously used this concentration of benzene in mixed-solvent MD simulations (Lama et al., Biochemistry, 2015). Further, from all seventeen peptides screened with the benzene probes, only one derivative, namely coreMYC (residues 101-150), underwent a significant ensemble shift, which clearly indicates that the influence of benzene on the conformational sampling of the different peptides is unbiased and determined by their sequence composition. As suggested by the reviewer, we have now added a time-series plot of R_g as a function of the simulation time in the revised version of the manuscript (Figure S6).

Row 182: Discussion on the enthalpic energy as favorable or unfavorable is misleading since the entropic contribution, which is a very significant contribution to the free energy in the extended conformations, is entirely ignored. Since the authors only rank the states based on the enthalpic energy and entirely dismiss entropy, the energy scores

are biased towards states bound to as many benzene molecules as possible, which is also presented in figure 1D. This ranking prevents the authors to rank the presence of “disorder-to-more-disorder” transitions, where extensive entropy contributes favorably to the free energy. The conclusion that the “high-energy extended states of coreMYC with exposed hydrophobic residues” is thus entirely misled by the chosen energetic ranking (enthalpy-based) and is therefore not supported by the data presented.

We observed from our MD simulations that the conformational ensemble of coreMYC (measured through R_g) is distributed between compact and extended states. The peptide is rich in bulky hydrophobic residues, whose exposure to solvent in the extended states will be energetically unfavourable and hence hydrophobic collapse will drive the formation of the compact states. Thus, our primary objective with this analysis was to develop a reaction coordinate which would provide an estimate of the energy associated with the different conformations of coreMYC. The computation of enthalpic energy through non-bonded interactions (van der Waals and electrostatics) and solvation energies (polar and non-polar solvation) should capture the conformational energy of the peptides; which we have done using the MM/GBSA method (Genheden et al., Expert Opin Drug Discov, 2015). As evident from the R_g -E plot, extended coreMYC is in the higher-energy states, while low-energy is only associated with compact states (Figure 1d). Thus, despite the lack of entropic calculation, our goal to get a description of the energy associated with the conformational states of the peptide is effectively met with this analysis and to this end, we believe that the conclusions drawn from the data is accurate. Further, the entropy which is generally computed using the normal mode analysis in MM/GBSA method is computationally very expensive, far from accurate and prone to large statistical error (Hou et al., JCI, 2011). Therefore, when absolute energy values are not critical, and the evaluation is primarily performed for relative energetic comparison, this term can be safely neglected as seen in other publications related to this methodology (Foloppe et al., Curr Med Chem, 2006; Homeyer et al., Mol Inform, 2012). In addition, benzene induces a shift in the conformational distribution of coreMYC by increasing the population of a subset of pre-existing configuration (towards the low R_g range), which signifies a “disorder-to-order” transition. Significantly, the lack of any new population or increase in the population of states towards the high R_g range indicates that it is not a “disorder-to-more-disorder” mode of ensemble modulation.

Conclusions from the experimental native mass spectrometry evaluation of EGCG binding to MYC appear biased towards the EGCG-bound state, which, according to Fig 2b, is only present in small amounts and at high excess of EGCG. Furthermore, the arrival time distribution is normalized and presented in an overlay with the unbound states, where the propensity of the longer arrival times can barely be discerned as being put behind the unbound sample, thereby graphically exaggerating the

propensity of the shorter arrival time. Notably, even with 2 EGCG molecules bound, the distribution rather appears to be 50-50 between closed (short arrival times) and “open” (longer arrival times), rather than a near-complete conversion suggested from MD simulations as emphasized in conclusions made by the authors. Given the relatively small signal for the EGCG-bound states, it would be relevant to show the errors in the distribution intensity for the 0, 1 and 2 EGCG molecules bound. The situation is similar for the data presented in figure 3c: only a very small portion of the TBP proteins are found to be in complex with MYC, thus weakening the conclusions that can be drawn from this experiments.

Native IM-MS is a well-established tool to monitor binding of small molecules to proteins and peptides. However, it requires transfer of the intact complexes to the gas phase, which can lead to loss of the interaction during a process termed “in-source dissociation”. As a result, peaks corresponding to unbound protein are observed in native MS even for high-affinity, specific complexes. To assess the degree of in-source dissociation, we performed direct titration experiments in ESI-MS. The maximum relative intensity of the protein:ligand complex under saturating conditions is the MS response, that is, the percentage of complex that can be preserved in native MS. The MS response observed here (around 0.4) is in the normal range for low-affinity protein-ligand complexes, see *e.g.* Fitzen et al., RCM 2009. We also observe a minor increase in compact states even for unbound protein (see new **Figure S8**) which indicates loss of bound ligands during desolvation.

For comparison, we examined EGCG binding to another disordered protein segment, the transactivation domain of p53, which we have studied previously (Kaldmäe et al., Structure 2021). This interaction has been verified extensively in solution and closely resembles the interaction between EGCG and coreMYC (Zhao et al., Nature Commun 2021). We find that both give a similar dose-dependent MS response.

Figure D. Native MS titration of coreMYC with EGCG shows a dose-dependent increase in 1:1 complexes between EGCG and coreMYC, with a maximum bound fraction of around 40% of the total protein. The p53 transactivation domain, a known non-specific binding partner for EGCG, exhibits a similar EGCG binding capacity.

To investigate whether the binding observed by MS reflects solution binding, we calculated the gas-phase K_d from the MS binding curve as well as from tryptophan fluorescence studies performed in solution. The K_d values are in good agreement with 155 and 149 mM, respectively, and in the same range as those reported for similar peptide-ligand pairs. This data is now included as **Figure S8**.

Furthermore, as we have pointed out in several places in the manuscript, the interaction between coreMYC and EGCG is not driven by occupation of a single specific binding site, but rather via non-specific association. As such, mixed populations of open and closed conformations for the coreMYC:EGCG complexes would be expected, depending on where the compound interacts with the peptide. This effect, together with the contribution from in-source dissociation, would lead to a shift from more open to more closed conformations (rather than a sudden, complete conversion) as the EGCG concentration increases, which is indeed what we observe (**Figure 2b-c**). We have plotted the arrival times as transparent, instead of filled, curves to better allow the reader to see the shift from predominantly extended to compact populations. We have also included arrival time plots for the 5+ charge state showing the same trend (**Figure S8c**). Perhaps most importantly, the NMR data (please see below) completely confirm EGCG binding and a significant shift to a more compact population. In summary, the MD, NMR, and IM-MS data show that one part of the total protein population is bound to EGCG and thus trapped in a more compact state, while another part is not bound and thus populates a broader conformational space.

Binding to MYC-box II follows the rules for “fuzzy” complexes, where finely tuned conformational landscapes are required for multimodal, stochastically driven interactions. While non-specific EGCG binding does not saturate the coreMYC population and thus does not elicit a complete conformational change, the binding is enough to shift the conformational landscape of coreMYC, which manifests itself in the loss of c-MYC-TBP as well as c-MYC-TRRAP interactions observed in cells. We now discuss these points in detail in the Discussion section of the revised manuscript.

The AF modeling cases (row 323 and onwards) are presented with no reporting of scores, in particular the iptm is lacking, which is the score for the interaction. Without these scores, it is impossible to judge the quality of the predicted structures (presented in Fig 3a,b, Fig S5). Furthermore, the MYC peptide docking to these structures in AlphaFold (using ref 31) is not based on the most recent multimer version of AlphaFold but instead uses the glycine linker “trick” to support formation of the complex in-silico, thereby limiting the number of attainable complex models and thus the conclusions

that can be drawn from these docking experiments. For an up-to-date and accurate benchmark on how to use peptide-docking for multimer this is the proper reference: 10.3389/fbinf.2022.959160, Johansson et al.

We would like to clarify that we have indeed used the multimer version (AlphaFold-multimer-v2) of AF2 in our modelling of coreMYC with the interacting protein partners. Our choice of reference (Tsuban et al., Nat Comm, 2022) supporting AF2 as a tool for protein: peptide docking have used the glycine linker as a means to generate docked complexes, which might have led to the confusion. We thank the reviewer for bringing the more appropriate reference (Johansson-Åkhe et al., Front Bioinform, 2022) to our attention, and we have now added this in the revised version of the manuscript.

Finally, with the exception of the modelling of the coreMYC:TRRAP and coreMYC:TBP complexes, the manuscript has no methods section pertaining to the AF modeling part of the work, nor any description of the modelled complexes presented in Suppl Fig 5, their similarity to already determined structures of the target proteins, or if they were modelled in their entity. Taken together, this lack of evaluation info and inappropriate approaches severely limit this section of the work as presented in its current state.

We have now performed a benchmarking study by predicting the structures of all experimentally determined structures of c-MYC in complex with its binding partners using AF2 (**Figure E**). The interface predicted TM (ipTM) scores of these complexes varies from 0.3-0.9. Models with higher ipTM scores (WDR5, TBP, and MAX) have a high degree of similarity, while models with lower ipTM scores (BIN1 and PNUTS) have variability in the binding mode between experimental and AF2 structures. However and more importantly in the context of our study, the conformation (measured in R_g) of the interacting c-MYC peptides show a good agreement for all complexes between experimental and AF2 structures. The ipTM scores for all the AF2 generated models of coreMYC with its binding partners varies between 0.3-0.5. Despite the lower scores for these complexes, our benchmarking exercise suggests that structural information on the conformation of coreMYC can be reliably extracted from the modelled complexes. We have now included this result in the revised manuscript (page 15 and **Figure S14**).

The overall aim from this part of the work was to examine the conformation of the bound state of coreMYC with known interactors (Table S1, Tu et al., BBA, 2015). The R_g of coreMYC as measured from these AF2 generated structures show that the peptide generally adopts an extended conformation in order to recognize its binding partners; which enabled us to postulate that the extended conformation is the active state of coreMYC, while the collapsed state is the inactive conformation. This was experimentally verified using native mass-spectrometry by evaluating the interaction of coreMYC with TRRAP and TBP before and after treatment with EGCG. We apologise for the lack of information on the methodology for which we have now added a

separate segment under the title “modelling with AlphaFold2” in the Methods section of the revised manuscript.

Figure E. Structure prediction of c-MYC: protein complexes by AF2. (a-e) Superimposition of experimentally resolved (blue *versus* green) c-MYC complexes with the top-ranked AF2 models (grey *versus* orange). The binding partners of c-MYC, PDB codes of the experimental structures, and the ipTM scores from AF2 of the different complexes are indicated. (f) Radius of gyration of c-MYC in the complexes computed from experimental (green) and AF2 (orange) structures.

In *cellulo*, the authors only show interruption of the coreMYC:TBP interaction, not the interruption of coreMYC:TRRAP. Given that the work presents analysis of these two interactions both by MD, AF and ESI, the coreMYC:TRRAP interaction should also be probed in cells. Furthermore, it was shown in Wei et al (Nature SMB 2019) that the interaction critical for tumor formation is not hydrophobic, but electrostatic, and focused on the convex side of the MYC:TBP complex instead of the amphiphatic helix bound to the concave, inner side of TBP. This renders the MYC:TBP interaction a peculiar biological control in the light of MYC inhibition model and therapeutic strategy suggested by the authors.

We agree with the reviewer on the need of analyzing MYC:TRRAP interaction *in cellulo* to make our study more robust. Thus, we have now included isPLA data demonstrating a decrease in MYC:TRRAP complexes upon EGCG treatment in SH-SY5Y cells (**Figures 4 and S22**). This result supports our findings and complements the isPLA data on MYC:TBP already presented in the previous version of the manuscript.

In Figure 4, the concentration of EGCG used in the MYC:TBP PLA experiment (25 μ M) exceeds the concentration where 20% of the cells are already showing reduced viability (Fig S10). Furthermore, the control experiment made to show dose-dependent reduction of nuclear MYC-TBP complexes is performed at even higher EGCG concentrations (50 and 100 μ M), at which the viability of the cells is decreased to below 50%, which surely must affect the outcome of the experiment. Furthermore, the obvious control experiment treating the MYC-MAX complex similarly with EGCG as the MYC-TBP complex to investigate specificity is to my understanding not done or not shown. The apparent presence of significant amounts of TBP in the cytosol is bewildering. Together, this adds unreliability to the cellular conclusions in this work.

We thank the reviewer for this critical comment. As WST-1 is a metabolic-based assay, and EGCG is described to affect cellular metabolism (PMID: 16476731; PMID: 34836374), we now have analyzed its dose-dependent effects on proliferation by live-cell imaging (IncuCyte, Essen Bioscience Inc., Ann Arbor, MI, USA) Germany) to complement the WST-1 viability assay presented in the previous version of the manuscript. Our results show that treatment with 25 μ M EGCG does not affect the survival or viability of the cells, while higher concentrations resulted in reduced viability. The new data, included in **Figure S21**, thus demonstrates that the cells well tolerate treatment with 25 μ M EGCG and we have thus used this concentration for the isPLA.

In addition, we have performed isPLA of MYC:MAX following EGCG treatment. Our results demonstrate that the MYC:MAX signals do not change upon increasing EGCG concentrations. This new data is shown in **Figure S22c**.

Regarding the cytosolic signals of TBP complexes, we would like to clarify that we only quantified the nuclear PLA signals. It has been reported that there is a small cytosolic pool of TBP, which furthermore engages in native interactions with other cytosolic proteins, see Pemberton et al., J Cell Biol. 145 1999. It is therefore not surprising that we can detect some of the protein in the cytosol.

Short comments:

Ref 27 should rather be Ref 28.

We have revised the numbering of the reference.

Figure S1 should be a line plot (as in Fig 1c) since curves behind the front data display are concealed and thereby cannot be evaluated or discussed.

The Figure has been updated as suggested by the reviewer.

- Do these prohibit publication or require revision

As of now, and as described above, this work has many features of a preliminary report and will need to be significantly strengthened in methodological, experimental and strategic aspects as well as in in-depth understanding of IDPs and their interactions. Current methodological progress in the IDP field needs to be considered, as well as reflection on concepts such as folding-on-binding versus conformational selection and entropy as a driving force for interaction. Furthermore, the study is still limited by the small number of small molecules assayed (only two) and the limited number of methods used.

As described above, we have now (a) reinforced the MD results with eight overlapping peptides and three additional molecular probes, (b) included additional MS and tryptophan fluorescence data to extract relative binding affinities of EGCG for coreMYC, (c) confirmed both EGCG binding to MYC box II and the resulting compaction of coreMYC by NMR spectroscopy (d) confirmed the effect of EGCG on the TRRAP interaction with isPLA data, and (d) showed that EGCG does not affect protein levels of MYC, MAX, TRRAP, or TBP nor *c-MYC* mRNA expression. Collectively, these additional data from computational, biophysical and cellular studies unequivocally support our initial interpretations. Further, our aim was to show proof-of-principle and not extending on showing a large number of interactions (*e.g.* a drug design approach based on our findings), which would be beyond the scope of the manuscript.

- Is the methodology sound? Does the work meet the expected standards in your field?

The IDP field has developed with incredible speed during the past couple of years, with lots of experimental approaches tailored to the analysis of IDP interactions. For a complex system such as this and given the extensive in-silico part of the investigation, at least one more biophysical method would be required to currently meet the expected standards in the IDP field. Normally this would be NMR, but it could also be a fluorescence method if solubility and/or access of the peptides/proteins is low. Note that no experiments in this work directly address affinity which leaves a methodological void.

In addition to the new cell biology, MD, and MS data, we have followed the reviewer's advice and performed extensive NMR experiments to validate our observations at a molecular level in solution. Specifically, we have:

- (a) analyzed the secondary structure of coreMYC and identified three transiently structured regions. We found that MBII, the functionally important part of coreMYC, is unstructured in solution, and that the MD trajectories accurately capture even transient structural features of coreMYC.
- (b) validated the interaction with EGCG by monitoring chemical shift changes during titrations. Titrations revealed non-specific association, in agreement with the new MS and tryptophane fluorescence titration data.
- (c) identified interacting regions in coreMYC that are most affected by EGCG binding. Chemical shift changes show that the amino-terminal half of MBII is most affected by EGCG binding, in agreement with the MD models.
- (d) detected the conformational change of coreMYC upon EGCG binding. Addition of saturating amounts of EGCG induces a significant reduction in the hydrodynamic radius of coreMYC, indicating the emergence of a compact population.

The NMR data are now included in the revised **Figure 2**, as well as in Supplementary **Figure S11**. We are grateful to the reviewer for the suggestion to include NMR, as the results have robustly strengthened our initial conclusions from MD and IM-MS and yielded additional information by identifying the preferred binding sites for EGCG on coreMYC.

- Is there enough detail provided in the methods for the work to be reproduced?

As described in my detailed review above, in several instances detail or even entire sections in methods are lacking, which in several instances precludes reproducibility.

We have provided a more detailed Methods section in our revised manuscript.

Reviewer #2 (Remarks to the Author)

High quality study described in a well crafted manuscript. In this study, the authors used molecular dynamic (MD) simulations and mass spectrometry to identify a conformational switch region of MYC, termed coreMYC. Indeed, MD simulations initially enable them to identify a core region of the intrinsically disordered MYC protein that displays a different degree of compactness in presence of water or benzene molecules. They then proceed to characterise the conformational landscape of this region using recombinantly produced 50 aa peptide and native mass spectrometry, in absence and presence of EGCG and to model the bound complexes of coreMYC to known MYC interactor proteins. Their results suggest that the more

high-energy extended states of coreMYC constitutes an active recognition module, while the compact conformation represents an inactive state of this peptide portion of MYC. Using in situ PLA, they observe a reduction of c-MYC:TBP complexes in neuroblastoma cells in presence of EGCG. Findings are novel and the work opens significant and exciting avenues for drugging the IDP region of MYC. The methodology is sound, and the work supports the conclusions and claims.

We are very happy for the positive feedback from the reviewer on our study.

Minor comment:

Does EGCG influences (reduces) MYC or TBP protein levels in SH-SY5Y cells? This appears like an important control to be able to conclude on the impact of EGCG on reducing the MYC:TBP complexes observed by isPLA: is the reduction of those complexes upon EGCG treatment due to the adoption of an inactive conformation of the switch or simply to a reduction of MYC protein levels? The authors should include a WB control for MYC protein levels before and after treatment with EGCG and ideally a qPCR assessing possible changes in MYC mRNA transcript.

We thank the reviewer for this comment and agree that checking the levels of the studied proteins after EGCG incubation is a crucial control. We have now performed Western blot analysis for protein levels. Cells were treated with 0, 10, 25, and 50 mM EGCG and protein expression analyzed after 24 hrs, *i.e.* the same conditions as used for the isPLA. As shown, incubation with 25 mM EGCG did not affect expression of any of the proteins used in this study, *i.e.* MYC, MAX, TBP, or TRRAP (Figure F; presented in S22d). In addition, we also analyzed *c-MYC* mRNA levels after incubation with 0, 10 and 25 mM EGCG. As shown, mRNA transcript levels did not decrease with 25 mM EGCG, as judged by RT-qPCR (Figure S22e). Thus, the reduction in PLA signals is due to inhibition of interaction between MYC:TBP and MYC:TRRAP and not due to reduced MYC levels.

Figure F. Effects of EGCG incubation on protein expression. (A) Western blot for MYC, TBP, MAX, and TRRAP from SH-SY5Y cells treated for 24 h with increasing concentrations of EGCG as indicated. As protein loading controls, β -actin was used for MYC, TBP, and MAX, and α -tubulin for TRRAP. Molecular weight markers in kDa are indicated to the left of the blots. **(B)** Quantification of Western blot assays, represented as protein expression relative to loading control (β -actin or α -tubulin) and normalized against 0 μ M EGCG. Data is presented as mean +SD from three independent replicates. Statistical analysis: two-way ANOVA with * indicating $p < 0.05$.

Reviewer #3 (Remarks to the Author)

The manuscript by Lama and coworkers describes probing conformational landscape of human c-MYC transcription factor using cosolvent (probe-based) molecular dynamics (MD) simulations and protein modelling, with validation provided by native mass spectrometry (MS) and cell-based assays. Using this approach, the authors identified a conformational switch within the N-terminal transactivation domain of c-MYC, which has not been reported previously. This switch shifts between a closed (inactive), and an open (active) conformation of the protein. Furthermore, the authors showed how the conformational landscape of c-MYC can be shifted towards its inactive conformation by non-specific small molecule binding.

This is highly significant, since c-MYC is the most frequently deregulated oncogene across the majority of human cancers. As most of its sequence lacks of a stable tertiary structure, c-MYC has been deemed “undruggable” by conventional structure-guided drug discovery approaches. Although the idea of shifting the ensemble of intrinsically disordered proteins is not new (see e.g. Kriwacki et al., J. Am. Chem. Soc. 2017, 139, 39, 13692), any approach that tackles the “druggability” of c-MYC deserves attention. Induction of closed conformation impedes interactions of c-MYC with the endogenous binding partners such as TRRAP and TBP (TATA-box binding protein), which are crucial for the oncogenic activity of c-MYC. Small molecule inducers of closed conformation may therefore represent an alternative strategy for “drugging” c-MYC. While the results presented in the manuscript are very early-stage and unlikely to translate into therapeutics any time soon, they represent a valid proof-of-concept.

The manuscript is very well-written, supported by high quality figures and supplementary information, and I thoroughly enjoyed reading it. The work is novel, and the outcomes would be of interest to the medicinal chemistry and cancer research communities. I have some questions regarding the methodology of the simulations, and I would like to suggest changes which would improve the manuscript. I would suggest acceptance after addressing these questions and comments.

We are grateful for the reviewer's praise in recognition of our work as an important novel proof-of-principle study.

1. It should be clarified why 50 residues segments were selected for independent simulations: the number seems somewhat arbitrary.

We postulated that polypeptides of 50 residues will reduce the structural complexity for conformational sampling of the disordered c-MYC protein, while still being adequate to efficiently represent the sequence-based dynamic and structural heterogeneity of the different regions of the protein. Further, the length was also defined by consideration of the number of peptide derivatives and system size that can be effectively subjected to long time-scale (microsecond range) all-atom explicit solvent MD simulations. Thus, it was a combination of an optimum stretch that can independently constitute the folding landscape of the various derivatives and the associated computational cost, that directed the empirical designing of 50 residue segments in the simulations and screening of probe-sensitive epitope within c-MYC.

2. While most of c-MYC is intrinsically disordered, some of its segments show elements of secondary/tertiary structure, which were captured by structural biology approaches (e.g. bHLH). As I understand, all 50-residues segments were modelled in fully extended polypeptide conformations and subjected to MD simulations. While radii of gyration plots (Fig. 1) show various degrees of compactness acquired during the simulation, there is no comparison between the most-populated clusters obtained during MD and those few segments that were solved by X-ray crystallography and NMR (e.g. PDB codes 2A93, 5I4Z). It would be helpful to show the secondary structure plots (e.g. DSSP) for those regions.

The initial structures of all the 50-residue peptide segments were indeed modelled in fully extended conformations and subjected to MD simulations. The two peptide derivatives Pep_8 (c-MYC₃₅₁₋₄₀₀) and Pep_9 (c-MYC₄₀₁₋₄₃₉) together corresponds to the basic Helix-Loop-Helix Leucine Zipper (bHLHZip) domain of c-MYC, whose structure has been experimentally resolved in complex with its binding partner MAX (e.g. PDB IDs: 2A93, 1NKP, 6G6K among others). As suggested by the reviewer, we have now analyzed the secondary structure evolution of these two peptides, which show significant presence of helical elements (**Figures Ga-b**). In Pep_8, residues 361-370 and 392-396 adopt a helical backbone geometry in more than 50% of the structures (**Figure Gc**), while for Pep_9, a single pre-dominant helical segment is observed between residues 416-435 (**Figure Gd**). We have also performed structure-based clustering of the simulated conformations, and a representative structure from the most populated cluster was compared with the structurally equivalent regions determined from experiments (**Figures Ge-f**). The Helix-Loop-Helix (HLH) topology in Pep_8 and the helical structure of the Leucine Zipper (Zip) in Pep_9 show good alignment between the simulated and experimental states. Taken together, our simulations indicate that

the carboxy-terminal region (residues 351-439) of c-MYC form alpha helices in solution, and adopt a metastable conformation, which is pre-disposed towards the bound-state complex structure. We have now included this analysis in the revised manuscript (Supplementary text and Figure S2).

Figure G. Secondary structures of c-MYC peptides corresponding to bHLHZip domain. (a-b) Secondary structure evolution of c-MYC peptides (residues 351-400 and 401-439) as a function of the simulation time analysed using DSSP algorithm. Para: Parallel beta sheet, Anti: Anti-parallel beta sheet, 3-10: 3_{10} helix, Alpha: Alpha Helix, Pi: Pi Helix. (c-d) Residue-wise percentage helicity of the two peptides computed using "secstruct" command from the cpptraj module of AMBER 18. The reported helicity is a sum of the "3₁₀" and "alpha" helices of the individual residues. (e-f) Representative structures from the most populated clusters of the two c-MYC peptides (orange) superimposed with the structurally equivalent segments from the experimentally resolved bHLH and Zip domains of c-MYC (blue). The percentage of structures in both the clusters are indicated. Structure-based clustering of the MD generated conformations of the two peptides was performed using average-linkage algorithm with pairwise RMSD as a distance matrix.

3. Cosolvent (probe-based) MD simulations: benzene (0.4 M), which has been used as a cosolvent/probe, has been noted to undergo phase separation due to stacking

interactions in extended MD simulations in aqueous solvents. Production trajectories reported in the manuscript were 5 microseconds. Was any of the phase separation of the hydrophobic probe observed? There is no information about it, and I am curious how the phase separation have been avoided.

The reviewer's concern is well-noted, and we have carefully examined for the co-aggregation of benzene probes in our simulations. At 0.4M the benzene molecules were still soluble, and we did not find any significant phase separation. In fact we have previously used the same molar concentration of benzene in mixed-solvent MD simulations after careful calibration for aggregation and efficiency for cryptic pocket detection on a protein surface (Lama et al., *Biochemistry*, 2015). Importantly, of all the peptides that were screened with the benzene probes in the present study, only one derivative (residues 101-150) underwent a significant ensemble shift (**Figures 1b, S3b and S5**), which clearly indicate that the influence of benzene on the conformational sampling of the different peptides is unbiased and determined by their sequence composition. We have not added any repulsive potential between the benzene molecules, and the phase-separation was avoided primarily by running the simulations in a concentration at which benzene is water miscible and also sufficient to conditionally impact the ensemble distribution of the peptides.

4. Although ff19SB and OPC are a good choice for the force field and solvent models for IDPs, a comment in the discussion on the selection/choice of the most appropriate force field should be added.

A major concern with atomistic simulations of IDPs employing currently available protein force-field is their tendency to favour overly compact structures (Huang et al, *Curr Opin Struct Biol*, 2018; Shabane et al., *JCTC*, 2019). In this regard, we carried out an independent assessment of three different AMBER-based protein force-fields (ff14SB, ff14IDPSFF, and ff19SB) with two water models (TIP3P and OPC) to sample the conformational states of all the initially designed nine non-overlapping c-MYC peptides (**Figure 4**). We found that "ff14SB + TIP3P" (Maier et al., *JCTC*, 2015), which is one of the standard combination of force-field and water model used in AMBER derived simulations sampled only the collapsed states for all the nine peptides. The combination of ff14IDPSFF (Song et al., *JCIM*, 2017; a force-field tailored for IDPs) and TIP3P relatively improved the sampling property of the peptides. Significantly, the most recently developed AMBER force-field ff19SB (Tian et al., *JCTC*, 2020), which has been parametrized with OPC water model was able to emulate the highest structural diversity with exploration of extended states across the different peptides. Based on this evaluation, we have performed our simulations in the study with ff19SB force-field and OPC water model. We have now included this in the revised manuscript (as Supplementary text and **Figure S1**).

5. MD simulations: all production runs were performed in duplicates. How consistent

were the observables obtained in each replica, e.g. radii of gyration? Were the results showed in Fig. 2b the results of averaging, or the trajectories were concatenated/combined?

MD simulations were indeed performed in duplicates, and the shift in the ensemble distribution of coreMYC towards a compact conformation in the presence of EGCG was consistently observed in all the independent production runs (**Figures S9a-b and S10a-b**). The measured radius of gyration ranges between 5-15 Å with the mean at 12 Å for all the replicates. Further, molecular characterization of each simulated trajectory showed that coreMYC:EGCG complexation is predominantly driven by hydrophobic interactions (**Figures S9e-f and S10e-f**). However, the residues involved in the recognition overlap but are not identical between the replicates, which corroborates our finding from native mass-spectrometry that indicated a generic mode of association between coreMYC and EGCG. Overall, the simulations were reproducible in underscoring the observation that EGCG induces coreMYC compaction through non-specific hydrophobic interactions. The results in **Figure 2b** (now new **Figure 2g**) is not derived from concatenated trajectories, rather we have shown the data from one independent MD run as a representation for each system (one or two EGCG molecules). The details of individual replicate for both the systems is also shown explicitly in **Figures S9 and S10**.

6. My concern is that only one type of cosolvent (benzene) and only one type of molecular probe (EGCG) were used in the study. The results would be a lot more compelling if several different cosolvents (hydrophobic, hydrophilic, aromatic, aliphatic, etc) were employed. Likewise, IDP modulators other than EGCG (see e.g. Berkeley and Debelouchina, Chem Sci 2022 Nov 21;13(48):14226) should be considered. EGCG is fine as a tool compound for IDP modulation *in silico* and *in vitro*, but because of its low affinity, low selectivity, toxicity in concentrations over 20 micromolar (mentioned in the manuscript), and poor bioavailability, it is unlikely to be used as a lead. It might be that better suited alternatives for ensemble-shifting exist.

Taking note of the reviewer's comment, we have now subjected coreMYC to additional mixed-solvent simulations in the presence of three different molecular probes, which include propane (hydrophobic: aliphatic), methanol (hydrophilic: h-bond donor), and acetaldehyde (hydrophilic: h-bond acceptor) (**Figure B**). We find that propane shifts the structural ensemble of coreMYC towards a compact conformation as was observed for benzene (hydrophobic: aromatic), while the hydrophilic compounds (methanol and acetaldehyde) have no significant impact on the epitope. Propane induces the conformational transition by acting as a molecular "glue" (like benzene) in stabilizing the hydrophobic collapsed state of coreMYC. These mixed-solvent simulations with multiple and diverse chemical fragments further emphasize the distinct sequence composition of this epitope, and its potential for conformational modulation by

exogenous ligands with specific pharmacophoric properties. We have now added this result in the revised manuscript (page 7 and **Figure S7**).

The selection of EGCG as a conformational modulator of coreMYC was based on the fact that it is a polyphenolic compound, which seemed like a natural extension of the benzene probes with better solubility for experimental validation. Besides, it has also been recently shown to interact with the disordered transactivation domain of the tumor suppressor p53 and to induce a compact conformation, which prevented its interaction with MDM2 (Zhao et al., Nat Comm, 2021). We agree that alternative ligands could be considered as modulators of coreMYC, and in that we thank the reviewer for directing us to the appropriate reference (Berkeley and Debelouchina Chem Sci, 2022). We also concur that EGCG itself would not be appropriate as a lead compound for drug development. However, our study (along with the work by Zhao et al., Nature Comm, 2021) provide a proof-of-principle that EGCG can provide the basic scaffold for lead identification and optimization that can be undertaken rationally based on the physicochemical nature of the ligand binding epitope in the target protein. The utility of such an approach has in-fact been recently demonstrated in the successful screening and identification of small molecules using EGCG-based pharmacophore that disaggregates AD-tau fibrils (Seidler et al., Nat Comm, 2022).

7. ECGC binding modes: although it is likely that ECGC exerts its effects via non-specific interactions, it would be recommended to further assess the predicted binding modes by running several shorter simulations (50 – 100 ns) with different starting coordinates of ECGC and monitoring the convergence of the binding poses showed in Fig. 2. An addition of molecular docking (using the closed conformation of coreMYC segment as a target) would also be beneficial in characterisation of those interactions.

We have now performed 2D NMR titration experiments, and the broad chemical shift perturbations observed in coreMYC with the addition of EGCG further affirms the non-specific nature of this peptide: ligand interaction (**Figure 2f**). Importantly, the residues which undergo significant chemical shifts are also observed to be predominantly engaged in contacting EGCG in the simulated ensembles. Clustering based on these residues provide the representative binding modes of EGCG and coreMYC (**Figures S9g-h and S10g-h**). We have further assessed the stability of these complexes by running multiple short simulations (100 ns each). The computed Root Mean Square Deviation (RMSD) shows that EGCG remains bound to coreMYC in all the simulation runs, although in the systems with two EGCG, there is relatively higher deviation due to reorientation of one of the bound ligands (**Figure H**). Collectively, this indicate that the proposed binding modes are potential representatives of the multiple metastable complexes that would be formed during the dynamic interaction between EGCG and coreMYC. We have now added this result in the revised manuscript (page 12 and **Figure S13**).

Figure H. Stability of EGCG bound to coreMYC. (a-d) Time-dependent Root Mean Square Deviation (RMSD) of EGCG for five independent simulations of systems with one (EGCG_1) or two (EGCG_2) EGCG molecules bound to coreMYC. The representative binding modes (termed Mode1 and Mode2) of coreMYC: EGCG complex (Figures S9g-h and S10g-h) were used as starting structures for these simulations. The RMSD was measured with reference to the starting structure for the respective systems.

Reviewer #4 (Remarks to the Author):

In this manuscript Lama et al describe the use of strategy combining computer simulations native mass spectrometry and cell biology to understand the conformational state(s) adopted by the intrinsically disordered protein c-Myc. They authors show how these conformations can be modulated within a precise region of c-Myc by small molecules and that this has consequences on the ability of c-Myc to participate in protein interactions. This is an elegantly designed and conducted study and the manuscript describes the data very well indeed. Here I highlight a few minor areas that the authors may wish to comment on when revising their manuscript for publication, focusing on the mass spectrometry elements of the work.

We thank the reviewer for the positive feedback on our study.

- Line 245 – The authors state that R_g is analogous to CCS. They should, however, cite some literature here discussing possibilities for protein compaction and other gas phase structural rearrangement phenomena that must be considered when interpreting ion mobility data.

This is indeed an important point, as folded proteins are known to undergo artificial compaction in the gas phase due to the collapse of *e.g.* unsupported loops (Rolland & Prell, Trends Anal Chem 2019), while disordered proteins can experience artificial compaction at low ion charge (Beveridge et al., Proteomics 2015). We have added the corresponding references to the section, to make the reader aware that R_g and CCS are not identical.

- Line 250 – The authors describe that the data show three major populations. However, on close inspection it looks as if the multimodal arrival time distribution could be fitted to at least 4 conformers (represented by Gaussian distributions). Are the authors calling the species between 2-4 ms one compact state? If so, the authors should consider adding labels to Figure 4c that splits the ATDs into these three states.

We agree that there are likely more sub-populations, likely due to slight differences in EGCG binding. We have clarified in the text on page 9 that we refer to the two closely neighbouring distributions (Figure 2c) with the lowest cross-sections as “compact states”.

- Are similar structural effects observed for the 5+ charge state of coreMYC? Even if not, the authors should report CCS values for this species and show that structural changes can or cannot be detected upon EGCG binding.

We indeed observe a compaction of the 5+ charge state as well, albeit to a lower extent. We speculate that the relatively high charge induces Coulombic stretching, making it harder to acquire any compact conformations. We have added the corresponding plots as **Figure S8c**.

- Line 269. How were the compact states of coreMYC selected from the MD simulations and what method/software was used to calculate the theoretical CCS? Are the experimental CCSs calculated from the peak top arrival time values? The authors should also check that their reporting of the ion mobility data are compliant with recommendations set out in PMID: 30707468.

We thank the reviewer for pointing this out and have changed to notation to the commonly used system (${}^TWCCS_{N2}$) and included the relevant parameters in the Methods section. CCS values were calculated using IMPACT. We apologize for the omission and have added this information to the methods section.

- Have the authors checked to see if EGCG binding modulates the conformations of full length c-Myc by ion mobility MS?

We have explored this idea, but since FL-c-MYC is completely disordered, we assume it will be difficult to specifically detect the effects of EGCG-induced local compaction around the coreMYC region. It is, however, an interesting possibility which we plan to investigate in the near future.

- Line 356. The authors conclude that EGCG inhibits the ability of coreMYC to interact with TRRAP. Does EGCG bind to TRRAP? How do the authors know that the inhibition is due to binding Myc and not TRRAP? A control experiment should be performed to check this and the language amended accordingly.

The reviewer is absolutely right that we cannot exclude EGCG affects complex formation by (additionally) blocking the binding site on TRRAP, especially since coreMYC and TRRAP are similarly hydrophobic. However, we observe that EGCG also blocks binding of coreMYC to TBP, which is a folded protein, and thus not subject to EGCG-induced compaction. These findings confirm that coreMYC compaction is in principle sufficient to abolish complex formation. To further clarify this point, we have now performed isPLA assays which confirm inhibition of the coreMYC-TRRAP interaction by EGCG. An investigation whether TRRAP, which is >3500 residues long and largely disordered, responds to EGCG binding, is certainly interesting, but outside of the scope of the present study.

- Figure 3c. Why is only one charge state detected for the coreMYC+TRRAP complex? Could the authors please comment on this.

We apologize for the omission, we can indeed detect three charge states for the coreMYC-TRRAP complex. We have added the zoomed spectrum as Supplementary Figure S16.

Figure I. Zoomed mass spectrum showing three charge states for the TRRAP-coreMYC complex. TRRAP is marked by blue and coreMYC by orange circles, respectively.

References

Fladvad M, Zhou K, Moshref A, Pursglove S, Säfsten P, Sunnerhagen M. N and C-terminal sub-regions in the c-Myc transactivation region and their joint role in creating versatility in folding and binding. *J Mol Biol.* 346, 175-89 (2005).

Tu WB, Helander S, Pilstål R, Hickman KA, Lourenco C, Jurisica I, Raught B, Wallner B, Sunnerhagen M, Penn LZ. Myc and its interactors take shape. *Biochim Biophys Acta.* 1849, 469-83 (2015).

Kalkat M, Resetca D, Lourenco C, Chan PK, Wei Y, Shiah YJ, Vitkin N, Tong Y, Sunnerhagen M, Done SJ, Boutros PC, Raught B, Penn LZ. MYC Protein Interactome Profiling Reveals Functionally Distinct Regions that Cooperate to Drive Tumorigenesis. *Mol Cell.* 72, 836-848.e7 (2018).

McMahon SB, Van Buskirk HA, Dugan KA, Copeland TD, Cole MD. The novel ATM-related protein TRRAP is an essential cofactor for the c-Myc and E2F oncoproteins. *Cell.* 94, 363-74 (1998).

Feris EJ, Hinds JW, Cole MD. Formation of a structurally-stable conformation by the intrinsically disordered MYC:TRRAP complex. *PLoS One.* 14, e0225784 (2019).

Huang J, MacKerell AD Jr. Force field development and simulations of intrinsically disordered proteins. *Curr Opin Struct Biol.* 48, 40-48 (2018).

Shabane PS, Izadi S, Onufriev AV. General Purpose Water Model Can Improve Atomistic Simulations of Intrinsically Disordered Proteins. *J Chem Theory Comput.* 15, 2620-2634 (2019).

Maier JA, Martinez C, Kasavajhala K, Wickstrom L, Hauser KE, Simmerling C. ff14SB: Improving the Accuracy of Protein Side Chain and Backbone Parameters from ff99SB. *J Chem Theory Comput.* 11, 3696-713 (2015).

Song D, Luo R, Chen HF. The IDP-Specific Force Field ff14IDPSFF Improves the Conformer Sampling of Intrinsically Disordered Proteins. *J Chem Inf Model.* 57, 1166-1178 (2017).

Tian C, Kasavajhala K, Belfon KAA, Raguetta L, Huang H, Miguez AN, Bickel J, Wang Y, Pincay J, Wu Q, Simmerling C. ff19SB: Amino-Acid-Specific Protein Backbone

Parameters Trained against Quantum Mechanics Energy Surfaces in Solution. *J Chem Theory Comput.* 16, 528-552 (2020).

Ghanakota P, Carlson HA. Driving Structure-Based Drug Discovery through Cosolvent Molecular Dynamics. *J Med Chem.* 59, 10383-10399 (2016).

Kimura SR, Hu HP, Ruvinsky AM, Sherman W, Favia AD. Deciphering Cryptic Binding Sites on Proteins by Mixed-Solvent Molecular Dynamics. *J Chem Inf Model.* 57, 1388-1401 (2017).

Lama D, Brown CJ, Lane DP, Verma CS. Gating by tryptophan 73 exposes a cryptic pocket at the protein-binding interface of the oncogenic eIF4E protein. *Biochemistry.* 54, 6535-44 (2015).

Genheden S, Ryde U. The MM/PBSA and MM/GBSA methods to estimate ligand-binding affinities. *Expert Opin Drug Discov.* 10, 449-61 (2015).

Hou T, Wang J, Li Y, Wang W. Assessing the performance of the MM/PBSA and MM/GBSA methods. 1. The accuracy of binding free energy calculations based on molecular dynamics simulations. *J Chem Inf Model.* 51, 69-82 (2011).

Foloppe N, Hubbard R. Towards predictive ligand design with free-energy based computational methods? *Curr Med Chem.* 13, 3583-608 (2006).

Homeyer N, Gohlke H. Free Energy Calculations by the Molecular Mechanics Poisson-Boltzmann Surface Area Method. *Mol Inform.* 31, 114-22 (2012).

Tsaban T, Varga JK, Avraham O, Ben-Aharon Z, Khramushin A, Schueler-Furman O. Harnessing protein folding neural networks for peptide-protein docking. *Nat Commun.* 13, 176 (2022).

Johansson-Åkhe I, Wallner B. Improving peptide-protein docking with AlphaFold-Multimer using forced sampling. *Front Bioinform.* 2, 959160 (2022).

Zhao J et al., EGCG binds intrinsically disordered N-terminal domain of p53 and disrupts p53-MDM2 interaction. *Nat Commun.* 12, 986 (2021).

Berkeley RF, Debelouchina GT. Chemical tools for study and modulation of biomolecular phase transitions. *Chem Sci.* 13, 14226-14245 (2022).

Seidler PM et al., Structure-based discovery of small molecules that disaggregate Alzheimer's disease tissue derived tau fibrils in vitro. *Nat Commun.* 13, 5451 (2022)

Rolland, AD, Prell, JS. Computational insights into compaction of gas-phase protein and protein complex ions in native ion mobility-mass spectrometry. *Trends Anal Chem* 116, 282-291 (2019)

Zhao et al., EGCG binds intrinsically disordered N-terminal domain of p53 and disrupts p53-MDM2 interaction. *Nature commun* 12, 986 (2021)

Kaldmäe et al., A "spindle and thread"-mechanism unblocks p53 translation by modulating N-terminal disorder. *Structure* 10, S0969-2126 (2022)

Beveridge et al., Relating gas phase to solution conformations: Lessons from disordered proteins. *Proteomics* 15, 2872-2883 (2015)

Fitzen et al., Peptide-binding specificity of the prosurfactant protein C Brichos domain analyzed by electrospray ionization mass spectrometry. *Rapid Commun Mass Spectrom* 23, 3591-8 (2009)

REVIEWER COMMENTS

Reviewer #1 (Remarks to the Author):

In the revised version of the manuscript, Dilraj Lama and co-authors have responded to several of the main points addressed earlier. They have extended their experimental and computational investigation using IM-MS, MD and cell biology, and they have expanded their peptide mapping thereby responding to unresolved issues in the earlier version. The methodology section has now also been expanded to cover previously undescribed parts of the work, such as the biostructural modeling and predictions using AF2. However, there are still parts of this manuscript that need to be addressed.

Line 192-202: Here, and in their response to the review, the authors continue to argue that entropy considerations can be omitted for this protein, which as the authors themselves realize by several techniques is intrinsically disordered to at least 80%. I do not agree with this – it simply does not make sense to talk about an extended, IDP state as a “high-energy state” just because entropy was not explicitly taken into account for computational-practical reasons (with references to works from 2006 and 2012...). Are the extended states for the other peptides high-energy states too, then? Furthermore, it is not even in agreement with their own data to describe the “compact” state (low Rg) as a low-energy state. In figure 1c, it is clear that the enthalpic energies of the compact, benzene-induced state cover the entire range of observed energy scores over a narrow range of Rg (approx. 3Å, the scale is unclear); furthermore, the core-MYC in water has similar Rg and energy distribution for the lower-Rg states as has the MYC in benzene. It is therefore not consistent with data to describe the “compact” state as a low-energy state since many of the highest energy states are also observed to have similarly low Rgs.

Line 289-296: The Kd obtained from fluorescence titration lacks error bars – and given the experimental data shown in Figure S8b this Kd is entirely unreliable as the binding curve could – within error - just as well be a straight line. With this data, claims of small-molecule binding to MYC in solution is not supported.

Line 316-345: There are some major flaws in how the NMR data is interpreted and what conclusions are being drawn from this. The reliability of the NMR results is not possible to judge without an HSQC spectrum showing the quality of the data obtained; good publication standards should include at least HSQC spectra with and without ligand, highlighting examples of smaller and larger CSPs (this could be in supplementary). Please note that the SSP analysis does not by itself provide evidence whether the fraction of secondary structure is “transient”. On line 333-34, it is stated that the average CSP is 0.06 ppm, but in figure 2f the average CSP is indicated as 0.11. It is unclear as to where the +sigma (+1 sigma?) calculation comes from – which average is used here? There is no description in Methods. On line 335 it is stated as important that the CSPs are larger when directly adjacent to Trp-135, however since this is the only Trp in this fragment and since small alterations of the orientation of an aromatic residue leads to larger changes in chemical shift solely based on ring currents, a direct implication for this region/residues to be involved in direct interactions cannot be made. Rather, since CSPs are observed in nearly the entire MYC100-150 fragment, it is more likely that the observed CSPs represent

an average over an ensemble of conformations in rapid conformational exchange, possibly interacting with EGCG but also just as well showing dynamic intramolecular interactions possibly induced by the presence of EGCG, which could possibly be in agreement with the MD data. The very small but possibly significant compaction of MYC in the presence of EGCG does not agree with the proposed transition from extended to “collapsed” which should be commented on. Taken together, the NMR data in solution only very weakly supports the hypothesis set up by the authors.

Other issues:

The authors use the terminology “IM-MS”, “IM” and “MS” in a mixed fashion that makes me confused as to whether different variants of the MS approach was used. Please be as consistent as possible when using these abbreviations and if one method is used without the other, then clarify.

The peptides: Throughout the manuscript, the peptides are described as “Pep_1 to Pep_9, Pep_10 to Pep_17”. This makes for a troublesome read and the Authors indeed often need to clarify the lab names in the text (see for example line 204). It would help the manuscript if the peptides were renamed by sequence, as MYC75-104 etc, providing immediate information to the reader.

Figure S14: The title of this figure should rather be “Structural validation of AF2 predictions of c-MYC complexes”, since these structures have already been experimentally determined and need not be predicted.

Figure S15: Here, the title can be “Structure prediction of coreMYC:protein complexes”.

Figure 5: To me, this schematic does not seem to agree with the presentation in the paper. Where is the extended, “active” but unbound conformation? Also, the main features of the modelled TRRAP and TBP complexes (Figure S15) are inconsistent with the ones presented in figure 5. In particular, the TRRAP-coreMYC complex modelled in figure S15 is predominantly showing a helical interaction surface, whereas in Figure 5 the interaction surface is extended; the same goes for the MYC-TBP complex.

Reviewer #2 (Remarks to the Author):

The authors well addressed my comments.

Reviewer #3 (Remarks to the Author):

I would like to thank authors for their detailed response to the comments raised and substantial improvements of the manuscript. The comments and revisions thoroughly addressed all concerns. I was

very pleased to see that authors included Figure G (DSSP plots), Figures S3b and S5 (indicating that the influence of benzene on the conformational sampling of the different peptides was unbiased), inclusion of the supplementary text and Figure S1, which addressed selection of the force fields, and revised section on p.7 and Figure S7 addressing ligand binding.

In addition to the new molecular dynamics simulations and extended data analysis, authors performed additional NMR experiments to validate their simulations at a molecular level in solution. This is very much appreciated. Specifically, authors have:

- (i) Analysed the secondary structure of coreMYC and identified three transiently structured regions. They found that the functionally important part of coreMYC is unstructured in solution, and that the MD trajectories adequately captured transient structural features of coreMYC.
- (ii) Validated the interaction with EGCG by monitoring chemical shift changes during titrations. Titrations revealed non-specific association, in agreement with the new MS data and Trp fluorescence titration data.
- (iii) Identified interacting regions in coreMYC that were most affected by EGCG binding, which agreed with MD simulations. Consistently with the MD trajectories, the chemical shift changes showed that the N-terminal region of MBII was the most affected by EGCG binding.
- (iv) Detected the conformational change of coreMYC upon EGCG binding: addition of saturating amounts of EGCG induced a significant reduction in the hydrodynamic radius of coreMYC, indicating the emergence of a compact population.

I am happy with the revised version of the manuscript and I can recommend it to be published in Nature Communications.

Reviewer #4 (Remarks to the Author):

The authors have addressed all of my comments in this revised version of the manuscript.

We thank the reviewer for these valuable and constructive comments, which have helped to strengthen our manuscript. We have carefully considered and addressed all points raised by modifying the main text, Figures 2 and 5, and the Supplementary Information accordingly. Please, find below a point-by-point response.

Reviewer #1 (Remarks to the Author):

In the revised version of the manuscript, Dilraj Lama and co-authors have responded to several of the main points addressed earlier. They have extended their experimental and computational investigation using IM-MS, MD and cell biology, and they have expanded their peptide mapping thereby responding to unresolved issues in the earlier version. The methodology section has now also been expanded to cover previously undescribed parts of the work, such as the biostructural modeling and predictions using AF2. However, there are still parts of this manuscript that need to be addressed.

Answer: We are grateful for the reviewer's appreciation of the additional work performed to address the comments and suggestions from the referee.

Line 192-202: Here, and in their response to the review, the authors continue to argue that entropy considerations can be omitted for this protein, which as the authors themselves realize by several techniques is intrinsically disordered to at least 80%. I do not agree with this – it simply does not make sense to talk about an extended, IDP state as a “high-energy state” just because entropy was not explicitly taken into account for computational-practical reasons (with references to works from 2006 and 2012...). Are the extended states for the other peptides high-energy states too, then? Furthermore, it is not even in agreement with their own data to describe the “compact” state (low R_g) as a low-energy state. In figure 1c, it is clear that the enthalpic energies of the compact, benzene-induced state cover the entire range of observed energy scores over a narrow range of R_g (approx. 3Å, the scale is unclear); furthermore, the core-MYC in water has similar R_g and energy distribution for the lower- R_g states as has the MYC in benzene. It is therefore not consistent with data to describe the “compact” state as a low-energy state since many of the highest energy states are also observed to have similarly low R_g s.

Answer: We agree with the referee that entropy is an important parameter to consider, but as outlined in detail in our earlier response, entropy calculations in this context would be computationally expensive and inaccurate. All-atom MD simulations as employed here are a suitable tool to assess the conformational preferences of peptides. The reviewer is correct that the compact states can populate both the low as well as the high enthalpic energy distribution. The extended conformations are almost exclusively associated with exposed hydrophobic residues, which results in generally unfavorable enthalpic energy. The compact states include conformations with favorable, as well as non-favorable intramolecular contacts. Benzene promotes compaction through hydrophobic interactions and not by globally lowering the energy of the peptide. We therefore agree that the terminology of “low and high energy states” was indeed inconsistent. We are grateful to the reviewer for bringing this to our attention and we have now removed any reference to the extended or compact states as “low” or “high” energy to reflect this.

Line 289-296: The K_d obtained from fluorescence titration lacks error bars – and given the experimental data shown in Figure S8b this K_d is entirely unreliable as the binding curve could – within error - just as well be a straight line. With this data, claims of small-molecule binding to MYC in solution is not supported.

Answer: We understand the reviewer's concern. The MS titration experiment suggests non-specific binding in the gas phase, which we want to confirm with an in-solution assay. Following the reviewer's advice, we have now removed the fluorescence data, and instead

added NMR data showing a linear increase in CSPs upon titration with EGCG (please see the new Figure S11c), which is a hallmark of non-specific binding. The NMR data thus also confirms the non-specific interaction between EGCG and coreMYC in solution and thus supports the observations from MD and MS, and we have adjusted the wording to better reflect this:

Line 316-345: There are some major flaws in how the NMR data is interpreted and what conclusions are being drawn from this. The reliability of the NMR results is not possible to judge without an HSQC spectrum showing the quality of the data obtained; good publication standards should include at least HSQC spectra with and without ligand, highlighting examples of smaller and larger CSPs (this could be in supplementary). Please note that the SSP analysis does not by itself provide evidence whether the fraction of secondary structure is “transient”.

Answer: We have now included the HSQC spectra with and without EGCG, as well as examples of peaks with small and large CSPs in the new Figure S11b. Please note that the raw data had already been made accessible in the BMRB with the previous submission. We furthermore concur that the SSP analysis may not be sufficient to determine whether a secondary structure motif exists only transiently and instead refer to the structures as “residual” to reflect this.

On line 333-34, it is stated that the average CSP is 0.06 ppm, but in figure 2f the average CSP is indicated as 0.11. It is unclear as to where the +sigma (+1 sigma?) calculation comes from – which average is used here? There is no description in Methods.

Answer: We thank the reviewer for bringing this to our attention. The value of 0.06, as stated in the text, is correct, while the value of 0.11 in the Figure was an error in the Y-axis formatting (*n.b.*, the graph is correct otherwise). We have corrected the axis and apologize for the mistake. The information how the sigma values were calculated has been added to legend of Figure 2.

On line 335 it is stated as important that the CSPs are larger when directly adjacent to Trp-135, however since this is the only Trp in this fragment and since small alterations of the orientation of an aromatic residue leads to larger changes in chemical shift solely based on ring currents, a direct implication for this region/residues to be involved in direct interactions cannot be made. Rather, since CSPs are observed in nearly the entire MYC100-150 fragment, it is more likely that the observed CSPs represent an average over an ensemble of conformations in rapid conformational exchange, possibly interacting with EGCG but also just as well showing dynamic intramolecular interactions possibly induced by the presence of EGCG, which could possibly be in agreement with the MD data.

Answer: The reviewer correctly points out that changes in Trp orientation can artificially increase CSPs, however, this would still require EGCG to induce structural changes in proximity of the Trp residue, which is what we proposed in the original manuscript. We have followed the reviewer’s advice by including the caveat that Trp can affect the CSP magnitude and toning down the likelihood of residues 127-131 being the major interaction site, as follows: *“The region spanning residues 127-131 exhibits the largest chemical shift changes in response to EGCG binding, although the magnitude of these changes may be affected by the proximity to Trp-135.”*

The reviewer is also correct that we observe CSPs (although of varying magnitude) across nearly the entire coreMYC sequence. The reviewer proposes as explanation that EGCG induces these CSPs by binding to coreMYC, or by increasing intramolecular interactions (which would also involve interactions with EGCG), both of which affect nearly all parts of the peptide. The reviewer’s explanation is entirely in line with our model: The association with EGCG is non-specific, as seen by MS and NMR titrations, and can occur at multiple sites,

although we maintain that some sites may be more favorable. These interactions with EGCG induce structural changes across the entire peptide, consistent with the global compaction evident from the MD simulations of coreMYC with one or two EGCG.

The very small but possibly significant compaction of MYC in the presence of EGCG does not agree with the proposed transition from extended to “collapsed” which should be commented on. Taken together, the NMR data in solution only very weakly supports the hypothesis set up by the authors.

Answer: The small but significant compaction observed by NMR, and its relation to the near-complete compaction seen in MD simulations, is explained in the Discussion section of the manuscript. The difference can be attributed to NMR being an ensemble method, where some of the peptides bind EGCG and become compact while others do not, whereas in MD, we sample individual molecules and observe a single state. To further clarify this point, we have now added a note to the end of the NMR results that the NMR and MD observations are compared in the Discussion section, as follows:

“It is important to note that this small compaction thus represents an ensemble value, which is compared to the significant compaction in MD simulations of individual coreMYC-EGCG in the Discussion section.”

Other issues:

The authors use the terminology “IM-MS”, “IM” and “MS” in a mixed fashion that makes me confused as to whether different variants of the MS approach was used. Please be as consistent as possible when using these abbreviations and if one method is used without the other, then clarify.

Answer: We apologize for the confusion and have now changed ion mobility MS to IM-MS, and native MS (without ion mobility spectroscopy) to nMS throughout the manuscript for consistency.

The peptides: Throughout the manuscript, the peptides are described as “Pep_1 to Pep_9, Pep_10 to Pep_17”. This makes for a troublesome read and the Authors indeed often need to clarify the lab names in the text (see for example line 204). It would help the manuscript if the peptides were renamed by sequence, as MYC75-104 etc, providing immediate information to the reader.

Answer: We agree that the numbering, which stems from the first version of the manuscript, where we only used 11 peptides, is impractical. We have followed the advice and now named the peptides with the actual residue numbers as suggested by the referee.

Figure S14: The title of this figure should rather be “Structural validation of AF2 predictions of c-MYC complexes”, since these structures have already been experimentally determined and need not be predicted. Figure S15: Here, the title can be “Structure prediction of coreMYC:protein complexes”.

Answer: These are excellent suggestions and have been implemented.

Figure 5: To me, this schematic does not seem to agree with the presentation in the paper. Where is the extended, “active” but unbound conformation? Also, the main features of the modelled TRRAP and TBP complexes (Figure S15) are inconsistent with the ones presented in figure 5. In particular, the TRRAP-coreMYC complex modelled in figure S15 is predominantly showing a helical interaction surface, whereas in Figure 5 the interaction surface is extended; the same goes for the MYC-TBP complex.

Answer: We agree that the Figure lacked consistency. We have updated the illustration to include an active, unbound state, as well as helical regions in the TRRAP and TBP complexes (please see new Figure 5). We believe that this Figure has become more balanced and thank the reviewer for the suggestion.

REVIEWER COMMENTS

Reviewer #1 (Remarks to the Author):

In this revision of the manuscript, the authors have responded and revised in a satisfactory way to a majority of my concerns. The entropy discussion is now revised, the lack-of-error-bars fluorescence titration has been removed (although it has value in indicating non-specific interactions), an explicit ensemble contextual interpretation has been included in the discussion section, and some other small items have been corrected (axis scaling, titles, readjustment of final model to include the disordered state).

On my request, the authors have now also included the NMR 2D HSQC data, from which the CSPs have been derived, in the Supplementary section. This data is however worrisome.

For an interaction leading to a more compact and thus more folded state, even in an ensemble of rapidly interconverting states, you would expect the NMR peaks to shift towards higher dispersion in the HSQC, “spreading out” the peaks in different directions. In contrast, in the titration of coreMYC with EGCG, the moving peaks systematically and clearly move in the same direction, ie towards higher ¹⁵N-shift and lower ¹H shift (“north-east” in the HSQC; Supplementary Figure 11C). This systematic effect is entirely concealed in Main Figure 2F, since the CSP calculation only provides the magnitude of the change, not the direction. Since the dispersion of the chemical shifts is not increased in adding EGCG, a structural change, or a shift in the conformational ensemble, is less likely to occur. Assuming DSS calibration and temperature control during the experiment has been performed correctly, this leaves us with systematic shifts of other kinds, such as salt-induced or pH-induced. EGCG is a weak acid which if affecting the pH during titration would lead to chemical shifts in the direction observed; indeed, the pH effect on ¹⁵N and ¹H shifts in Thr in the EGCG-MYC titration is particularly pronounced, as would be expected for such effects (Platzer et al. 2014, PMID: 25239571). Furthermore, it is not stated in the experimental section how the EGCG titrand was buffered – with such small CSPs, it would have to be in the same buffer as MYC to avoid general salt effects on the chemical shifts (see for example Kukic et al. 2013, Chem Phys Lett).

Taken together, in reviewing the complete NMR data set, I find it clearly NOT possible to deduce from these experiments that there are specific interactions between EGCG and coreMYC. The systematic effects in the HSQC spectra in Suppl Fig S11C are incredibly obvious and based on these spectra, interactions CANNOT be interpreted on a per-residue level. If there are issues with the calibration of the respective HSQCs, addressing this may resolve the issue and provide an improved picture. If not, I would say that the NMR titration with EGCG tells the exact same story as the fluorescence experiment: any possible binding of EGCG to MYC is unspecific. This does not exclude, however, that it may lead to a dynamic compaction of the conformational ensemble of coreMYC.

Suggested revisions:

Please check your calibration of the HSQCs carefully to remove any presence of systematic errors in the

NMR spectra. This may resolved your issues. If calibrations are correct, then re-do the NMR titration with careful control on pH and salt conditions, or revise as below.

The authors write:

Line 467-470: Further, from NMR and MD, we find that the the hydrophobic residues from coreMYC are locked in a pseudo-core with EGCG (Figures 2f;g; Figures S9e-h; S10e-h), which would make them unavailable for interactions with TRRAP2038-2087.

Suggested revision:

Take out NMR from this sentence, the data does not support this. There is no indication of any locking of residues as the experimental data stands, and the MD results surely suggest transient interactions rather than locked interactions with “unavailable” residues – so polish this writing too.

The authors write:

Line 666-668: Here, employing a combination of MD simulations, IM-MS, and NMR, we show that EGCG interacts at multiple sites with the hydrophobic residues present in the coreMYC peptide.

Suggested revision:

Either take out NMR, or take out “with the hydrophobic residues present”.

Reviewer #1 (Remarks to the Author):

In this revision of the manuscript, the authors have responded and revised in a satisfactory way to a majority of my concerns. The entropy discussion is now revised, the lack-of-error-bars fluorescence titration has been removed (although it has value in indicating non-specific interactions), an explicit ensemble contextual interpretation has been included in the discussion section, and some other small items have been corrected (axis scaling, titles, readjustment of final model to include the disordered state).

On my request, the authors have now also included the NMR 2D HSQC data, from which the CSPs have been derived, in the Supplementary section. This data is however worrisome. For an interaction leading to a more compact and thus more folded state, even in an ensemble of rapidly interconverting states, you would expect the NMR peaks to shift towards higher dispersion in the HSQC, “spreading out” the peaks in different directions. In contrast, in the titration of coreMYC with EGCG, the moving peaks systematically and clearly move in the same direction, ie towards higher ¹⁵N-shift and lower ¹H shift (“north-east” in the HSQC; Supplementary Figure 11C).

We understand the concern but disagree with the reviewer on these points. It is important to remember that we are not claiming that coreMYC folds upon binding EGCG. Our hypothesis is that interactions between coreMYC and EGCG result in a transiently compacted conformation. Although most peaks move “North-East” upon addition of EGCG, there is a clear sequential variation in how much the individual peaks move, which cannot be explained by a general effect. We also observe peaks that almost do not change their positions (including Thr-residues) and peaks that move in a substantially different direction compared to the other. For some peaks we even see substantial line-broadening suggesting slow exchange between two (or more) conformational states (we have added more magnifications to Figure S11C to highlight this). Furthermore, the signals from the Asn and Gln side-chains move very little compared to the signals from backbone amides, which demonstrates that the environment surrounding the backbone amides change more than the environment around the Asn and Gln sidechains. This behavior is consistent with a redistribution of the conformational ensemble of the polypeptide backbone rather than a general solvent effect.

This systematic effect is entirely concealed in Main Figure 2F, since the CSP calculation only provides the magnitude of the change, not the direction. Since the dispersion of the chemical shifts is not increased in adding EGCG, a structural change, or a shift in the conformational ensemble, is less likely to occur.

EGCG is known to affect the conformational ensemble of the IDPs α -synuclein, A β and p53, for which the changes in the NMR spectra are qualitatively similar (see e.g. references 28 and 29).

We have added the following to the presentation of the NMR titration on page 11 of the revised manuscript: “Most peaks are perturbed by the addition of EGCG. This effect of EGCG on coreMYC is similar to what has been observed in titrations of other IDPs with EGCG.^{28,29} We do however observe clear sequence dependent effects ranging from almost no change in chemical shifts to large changes where the lines are also broadened (Figure S11c).

Assuming DSS calibration and temperature control during the experiment has been performed correctly, this leaves us with systematic shifts of other kinds, such as salt-induced or pH-induced. EGCG is a weak acid which if affecting the pH during titration would lead to chemical shifts in the direction observed; indeed, the pH effect on ¹⁵N and ¹H shifts in Thr in the EGCG-MYC titration is particularly pronounced, as would be expected for such effects (Platzer et al. 2014, PMID: 25239571).

We thank the reviewer for suggesting the technical issues that could underlie the observed chemical shift changes. For each concentration of EGCG in the titration series, we calibrated the spectra to the frequency of DSS. We also measured the pH in the sample after the titration and found it to be 7.5 and thus unchanged by the addition of 1 mM EGCG. Furthermore, it is highly unlikely that the change in ionic strength resulting from the addition of 1 mM EGCG from a 10 mM stock dissolved in water would have any effect on the NMR spectra of a sample containing 10 mM HEPES, pH 7.5, 2 mM TCEP, 5% D₂O, 0.01 % NaN₃ and 0.25 mM 2,2-dimethyl-2-silapentane-5-sulfonic acid.

Although we have shown that the pH is constant throughout the titration series, it is unclear why pH effects should be particularly pronounced for Thr residues. The cited paper is concerned with pH effects on the random coil chemical shifts of ionizable amino acid residues (specifically, Asp, Glu, His, Cys, Tyr, Lys, and Arg). However, Thr103 is not ionizable unless it is phosphorylated, which is not the case in our sample. Also, the adjacent Thr123 displays virtually no shift at all (see Figure S11).

To emphasize that we both carefully calibrated the chemical shifts of each point in the titration series and measured pH before and after the titration we have revised the paragraph about the EGCG titration in the Methods section in the Supplementary Information, so it now reads: "A series of ¹⁵N-HSQC spectra were recorded with increasing EGCG concentrations from 0 mM to 1 mM added from a 10 mM stock of EGCG in water. pH of the sample was measured before and after the titration series and was unaffected by the addition of EGCG (pH = 7.5). For each spectrum in the titration series the chemical shift was calibrated to the frequency of DSS. As saturation was not reached under the NMR conditions, the effect of EGCG was calculated as the weighed change in ¹H and ¹⁵N chemical shifts between the lowest and highest EGCG concentration for each backbone amide group."

Furthermore, it is not stated in the experimental section how the EGCG titrand was buffered – with such small CSPs, it would have to be in the same buffer as MYC to avoid general salt effects on the chemical shifts (see for example Kukic et al. 2013, Chem Phys Lett).

We again thank the reviewer for suggesting a possible experimental pitfall. Our EGCG stock was prepared in pure water and the titration will thus lead to some dilution of the components in the samples. However, the e.g. the HEPES will be diluted by no more than 1 mM. The chemical shift changes analyzed in the paper by Kukic et al are for changes in NaCl concentrations of 50 – 200 mM, which is not comparable to the situation in this work. We have added to the methods section how the EGCG stock was prepared (see the above point).

Taken together, in reviewing the complete NMR data set, I find it clearly NOT possible to deduce from these experiments that there are specific interactions between EGCG and coreMYC. The systematic effects in the HSQC spectra in Suppl Fig S11C are incredibly obvious and based on these spectra, interactions CANNOT be interpreted on a per-residue level.

We are not suggesting the formation of long-lived specific interactions between coreMYC and EGCG. Rather we suggest that the interactions are transient and non-specific. We already mention this several times in the manuscript, e.g. on page 9, 12, 23, 24, 25, 26.

If there are issues with the calibration of the respective HSQCs, addressing this may resolve the issue and provide an improved picture.

As stated above the chemical shifts referencing have been rigorously performed at each EGCG concentration, there is no change in pH and the dilution effect from adding EGCG is vanishingly small.

If not, I would say that the NMR titration with EGCG tells the exact same story as the fluorescence experiment: any possible binding of EGCG to MYC is unspecific. This does not exclude, however, that it may lead to a dynamic compaction of the conformational ensemble of coreMYC.

We completely agree with the reviewer. Our conclusion is exactly that non-specific EGCG interactions leads to a dynamic compaction, which is supported by our MD simulations, MS and NMR experiments. We have revised the text to emphasize this conclusion (see below), and included an additional, explanatory statement in the results section:

“Most peaks are perturbed by the addition of EGCG. This effect of EGCG on coreMYC is similar to what has been observed in titrations of other IDPs with EGCG^{28,29}. We do however observe clear sequence dependent effects ranging from almost no change in chemical shifts to large changes where the lines are also broadened (Figure S11c).”

Suggested revisions:

Please check your calibration of the HSQCs carefully to remove any presence of systematic errors in the NMR spectra. This may resolved your issues. If calibrations are correct, then redo the NMR titration with careful control on pH and salt conditions, or revise as below.

As stated above the chemical shifts referencing have been rigorously performed at each EGCG concentration, there is no change in pH and the dilution effect from adding EGCG is vanishing.

The authors write:

Line 467-470: Further, from NMR and MD, we find that the the hydrophobic residues from coreMYC are locked in a pseudo-core with EGCG (Figures 2f;g; Figures S9e-h; S10e-h), which would make them unavailable for interactions with TRRAP₂₀₃₈₋₂₀₈₇.

Suggested revision:

Take out NMR from this sentence, the data does not support this. There is no indication of any locking of residues as the experimental data stands, and the MD results surely suggest transient interactions rather than locked interactions with “unavailable” residues – so polish this writing too.

We thank the reviewer for bringing the inaccurate wording of this sentence to our attention. We have revised the sentence which now reads: “Furthermore, from MD simulations and NMR, we conclude that coreMYC engages in nonspecific interactions with EGCG (Figures 2f-g; Figures S9e-h; S10e-h) that reduce its ability to interact with TRRAP₂₀₃₈₋₂₀₈₇.”

The authors write:

Line 666-668: Here, employing a combination of MD simulations, IM-MS, and NMR, we show that EGCG interacts at multiple sites with the hydrophobic residues present in the coreMYC peptide.

Suggested revision:

Either take out NMR, or take out “with the hydrophobic residues present”.

We have rewritten the sentence which now reads: “Here, employing a combination of MD simulations, IM-MS, and NMR, we show that EGCG interacts transiently with the coreMYC peptide.”

Figure S11. NMR analysis of coreMYC and coreMYC in the presence of EGCG. (a) HSQC spectrum of coreMYC in the absence of EGCG. **(b)** Secondary chemical shifts for $^{13}\text{C}\alpha$, $^{13}\text{C}'$, $^{13}\text{C}\beta$, and ^{15}N atoms. **(c)** HSQC spectrum overlay of coreMYC in the presence of increasing EGCG concentrations. Examples for residues that exhibit minor or major chemical shift perturbations (Ala150, Thr103, Thr1223, and Phe138, respectively) are shown as magnification. **(d)** Chemical shift perturbations for each residue at EGCG concentrations from 0 to 1 mM. Each residue number is indicated above each graph. **(e)** Diffusion of coreMYC. The intensity of methyl signals were measured as a function of the strength of the field gradients in the absence of EGCG (black triangles) and in the presence of 1 mM EGCG (orange circles). The straight lines are fits to the Stejskal-Tanner equation to obtain the diffusion constant³⁴.

Reviewers' Comments:

Reviewer #5 (Remarks to the Author):

In the manuscript by Lama et al. the authors use an interdisciplinary approach to characterise a small region of c-Myc (coreMYC; residue 101-150), which also includes the MBII site and a site that forms an amphipathic alpha-helix when bound to TBP. The authors suggest, based on MD simulations, NMR, and mass spectrometry that the coreMYC region becomes more compact upon interaction with EGCG.

As requested by the editor, I will mainly comment on the NMR characterisations and the comments made by another reviewer of this manuscript.

Overall, this manuscript shows a potential interesting aspect of region 101-150 of c-MYC. However, some of the results presented do appear preliminary in nature, where more validations of the approaches would have benefitted the manuscript.

I have a few points to be considered:

1) Reviewer #1 brings up a very valid point regarding the 2D NMR spectra shown in Figure S11. It is rather atypical to observe that a vast majority of peaks move in one direction and by a similar amount. The case that I know of are – change in pH, change in temperature, and from unfolded to folded transition, but where the peaks move from {small 1H ppm, large 15N ppm} -> {large 1H ppm, small 15N ppm}. As such, I do not believe that these spectra are convincing in showing the binding that the authors claim.

- a. The authors do show a change in the diffusion constant, which could indicate compaction (or change in viscosity).
- b. One way to robustly show the binding would be to record either CEST or CPMG experiments. At least one of the authors have published results using these experiments before and the samples required are the same as those already used.
- c. The authors could also follow the EGCG titration with ¹³C-1H spectra, where the 1H_{alpha} and ¹³C_{alpha} are excellent reporters on structural changes. These chemical shift changes can easily be compared with those calculated from the MD simulations to confirm agreement between the two methods.

2) The forcefield used in the MD simulations (ff19SB with the OPS water) has recently been shown to “produce overly compact conformational ensembles and show discrepancies in the secondary structure content compared to the experimental data” [ref: <https://pubmed.ncbi.nlm.nih.gov/35950933/>]. The authors do try to validate this force-field for their system, as described in supporting material. However, for any quantitative assessment of a force-field, one must benchmark against experimental data, e.g. SAXS or NMR chemical shift / scalar couplings, which has not been done.

3) The mass spectrometry also appears slightly preliminary. Specifically, it is well known that IDPs can

collapse during IM (check here from one of the IM-MS experts:
<https://pubmed.ncbi.nlm.nih.gov/29428839/>). By the way – it is mass spectrometry _not_ mass spectroscopy.

Please find enclosed a point-by-point response to the concerns raised by reviewer #5.

1) Reviewer #1 brings up a very valid point regarding the 2D NMR spectra shown in Figure S11. It is rather atypical to observe that a vast majority of peaks move in one direction and by a similar amount. The cases that I know of are – change in pH, change in temperature, and from unfolded to folded transition, but where the peaks move from {small 1H ppm, large 15N ppm} -> {large 1H ppm, small 15N ppm}. As such, I do not believe that these spectra are convincing in showing the binding that the authors claim.

a. The authors do show a change in the diffusion constant, which could indicate compaction (or change in viscosity).

b. One way to robustly show the binding would be to record either CEST or CPMG experiments. At least one of the authors have published results using these experiments before and the samples required are the same as those already used.

c. The authors could also follow the EGCG titration with ¹³C-¹H spectra, where the ¹H_α and ¹³C_α are excellent reporters on structural changes. These chemical shift changes can easily be compared with those calculated from the MD simulations to confirm agreement between the two methods.

We understand the reviewer's concern. As reviewer #1 and the previous handling editor [REDACTED] have stated, the NMR titration data is not relevant for the findings reported in this work. As the experiments suggested by reviewer #5 are beyond the scope of this work, we have therefore followed the advice from [REDACTED] and removed the NMR data (Figure 2e-f and Figure S11, the method section on NMR in the SI Appendix as well as the text on pages 10-11). See below for the modified Figure 2. We have furthermore added the following statement on page 10:

"We additionally sought to validate the effect of EGCG on coreMYC using solution NMR. However, the chemical shift perturbations observed in coreMYC during EGCG titrations could not be separated with sufficient confidence from potential titration artefacts. The data has therefore not been included in this study."

Figure 2. Characterization of the interaction between coreMYC and EGCG (a) The chemical structure of EGCG. (b) Native mass spectra of coreMYC in the presence of increasing concentrations of EGCG. The observed charged states and number of EGCG molecules bound to coreMYC are indicated. (c) Arrival time distributions for the 4+ charge state of coreMYC in different concentrations of EGCG. (d) R_g -E (energy normalized in the scale from 0 to 1) distribution plots of coreMYC generated from MD simulations of the peptide with one (upper panel) or two (lower panel) EGCG molecules (red), respectively. R_g -E distributions of coreMYC in water (black) is shown for reference. (e) Representative structures from MD simulations of coreMYC bound to one (upper panel) or two (lower panel) EGCG molecules, respectively. The residues with contact score > 1 are shown in spheres and the EGCG is depicted in stick representation (green). Side-chain carbon atoms of hydrophobic residues are shown in yellow and hydrophilic residues in orange. Oxygen and nitrogen atoms are shown in red and blue respectively.

2) The forcefield used in the MD simulations (ff19SB with the OPS water) has recently been shown to “produce overly compact conformational ensembles and show discrepancies in the secondary structure content compared to the experimental data” [ref: <https://pubmed.ncbi.nlm.nih.gov/35950933/>]. The authors do try to validate this force-field for their system, as described in supporting material. However, for any quantitative assessment of a force-field, one must benchmark against experimental data, e.g. SAXS or NMR chemical shift / scalar couplings, which has not been done.

3) The mass spectrometry also appears slightly preliminary. Specifically, it is well known that IDPs can collapse during IM (check here from one of the IM-MS experts: <https://pubmed.ncbi.nlm.nih.gov/29428839/>). By the way – it is mass spectrometry not mass spectroscopy.

The reviewer states that IDPs can appear overly compact in MD and in MS measurements. As described in detail in the text (page 5), Figure 1, and the associated Supplementary data (Figures S1-S3), these issues are the precise rationale of the reductionist approach we have employed in our study. In short, we analyzed **individual, overlapping peptides**, not the entire IDP, and monitored individual conformational changes, not systemic compaction. The experimental approach using peptides, MD, and MS, has been judged to be technically sound by reviewers #1, 2, 3, and 4, with reviewer #3 (who is an expert on MD simulations) specifically applauding our use and benchmarking of ff19SB-OPC.

Reviewer #5 states that the ff19SB-OPC force field causes artificial compaction of IDPs, based on a recent report using a single protein (the 170-residue IDP α Synuclein) as test case. We have specifically addressed this concern in our study as highlighted with two major points below.

1. The entire point of our reductionist peptide-based approach was to reduce the complexity for conformational sampling of the disordered c-MYC protein, while still being adequate to efficiently represent the sequence-based dynamic and structural heterogeneity of the different regions of the protein. We designed 50-residue overlapping peptide derivatives across the length of the IDP. We have then performed an independent assessment by exhaustively benchmarking the conformational sampling of these peptides against three force-fields: ff14SB (Maier et al., JCTC, 2015), ff14IDPSFF (Song et al., JCI, 2017), and ff19SB (Tian et al., JCTC, 2020), and two water models (TIP3P: a three-point water model, and OPC: a four-point water model). Using these assessments, we found that the combination ff19SB+OPC was able to emulate the highest structural diversity across the different peptides. Importantly, we want to emphasize that the peptide derivatives did not just sample the compact states. Instead, the different peptides exhibited variable sampling properties with significant population of extended states observed for many peptides (see Figure S1). This clearly indicates that the forcefield is not biased towards the sampling of only the compact states. Hence based on this evaluation, we performed our simulations with ff19SB force-field and the OPC water model. Further, in an independent study the combination of ff19SB+OPC has been shown to outperform the other force fields for modelling the structures of disordered *and* ordered peptides in the size range considered for our study (Abriata and Peraro, Comp Struct Biotech J, 2021). It has also been demonstrated that the four-point OPC water model in combination with popular amber-based protein force-field produces a more realistic representation of structural ensemble of multiple studied IDPs which are around 50 residues or shorter (Shabane et al, JCTC, 2019).
2. We agree that a direct benchmarking from simulations with experiments would enable a quantitative assessment. However, the specific purpose of the simulations was not to obtain absolute conformations, but instead to screen for an epitope within the

intrinsically disordered c-MYC protein whose structural ensemble can be modulated with exogenous ligands. For this, we compared the conformational sampling of the seventeen different c-MYC peptide derivatives in the presence and absence of benzene probes. A comparison across all seventeen derivatives highlights that the conformation of only one of the peptides (c-MYC₁₀₁₋₁₅₀) was significantly shifted towards a compact conformation by the addition of benzene (see Figures 1 and Figure S3). The other peptide derivatives were largely unaffected by the addition of benzene which clearly emphasizes that this is not a systemic effect. Rather, the distinct sequence composition of the c-MYC₁₀₁₋₁₅₀ makes it particularly susceptible to conformational modulation by benzene. Further, this MD observed ensemble modulation of the c-MYC peptide was validated with native MS and IM-MS using the polyphenolic compound EGCG, providing experimental support for the effectiveness of the designed MD methodology in mapping ligand inducible shape-shifting epitopes (see Figure 2 and Figures S8-10).

The reviewer states that IDPs can undergo compaction in the gas phase, citing work by Barran et al, *Curr Opin Chem Biol* 2018. This is absolutely correct, but as with the concerns about the MD simulations, our experimental strategy is specifically designed to address this problem.

1. As stated above, and in the manuscript, the very rationale of our reductionist approach is to avoid the complex issue of analyzing a full-length IDP, also with MS. IDPs populate both compacted and extended states in native MS due to simultaneous ionization following the charged residue model (CRM) and the chain ejection model (CEM). This leads to the well-known bimodal charge state distributions observed in MS of IDPs, where the lower charge state envelope represents overly compact states that stem from ionization according to the CRM, with $z(\text{max})$ far below the number of basic sites (see Borysik et al, *JACS* 2015, for an in-depth comparison of IDPs with MS, MD, and SAXS). However, the 50-residue segment analyzed here is a peptide, not a full-length IDP. Peptides ionize via the chain ejection and potentially ion evaporation models, which will produce extended rather than compact states (for all this, see the extensive bodies of work by Perdita Barran, Lars Konermann, Evan Williams, and others). Our data clearly confirms this already well-known fact, as we can detect a single charge state distribution, with $z(\text{max})=5$, as expected for CEM ionization. We observe multiple gas phase conformers by ion mobility, arguing strongly against the suggestion that the peptide had undergone any significant compaction. There is an extensive body of work showing that native IM-MS accurately reflects structures and interactions of disordered peptides in the same size range as coreMYC see *e.g.* extensive work by Michael Bowers, Kevin Pagel, Brandon Ruotolo, and others, who routinely combine IM-MS and MD of peptides (for example, Wyttenbach et al, *Annu Rev Phys Chem* 2014, Soper et al, *PCCP* 2013, Ilitchev et al, *JACS* 2018).
2. As with the MD simulations, we use the IM-MS data to detect conformational *changes*, not absolute conformations. We show that the coreMYC peptide becomes compact only *after* ligand binding within the same charge state. This fact directly contradicts the claim of reviewer #5 that we may observe indiscriminate gas phase compaction. We furthermore point to the positive assessment of our data and interpretations by reviewer #4, who is an IM-MS expert.

Lastly, it is worth to note that the secondary structure propensities measured by NMR, and which have not been questioned by any of the reviewers, match very well with the ones

observed by MD, and that the collision cross-sections measured by IM-MS agree with the values expected from MD, underscoring the validity of the entire MS+MD approach.

Furthermore, the reviewer makes the statement “By the way - it’s mass spectrometry_not_mass spectroscopy”. However, we have not used the term “mass spectroscopy” anywhere in the manuscript or in the Supplement Appendix. Maybe the reviewer refers to the term “ion mobility spectroscopy”, which was a typo which we now have corrected.